# Deep learning enables reference-free isotropic super-resolution for volumetric fluorescence microscopy

Hyoungjun Park [1], Myeongsu Na[2], Bumju Kim[3], Soohyun Park[4], Ki Hean Kim [3,4], Sunghoe Chang [2,5] & Jong Chul Ye [1,6 ✉]

Volumetric imaging by fluorescence microscopy is often limited by anisotropic spatial resolution, in which the axial resolution is inferior to the lateral resolution. To address this problem, we present a deep-learning-enabled unsupervised super-resolution technique that enhances anisotropic images in volumetric fluorescence microscopy. In contrast to the existing deep learning approaches that require matched high-resolution target images, our method greatly reduces the effort to be put into practice as the training of a network requires only a single 3D image stack, without a priori knowledge of the image formation process, registration of training data, or separate acquisition of target data. This is achieved based on the optimal transport-driven cycle-consistent generative adversarial network that learns from an unpaired matching between high-resolution 2D images in the lateral image plane and low-resolution 2D images in other planes. Using fluorescence confocal microscopy and light-sheet microscopy, we demonstrate that the trained network not only enhances axial resolution but also restores suppressed visual details between the imaging planes and removes imaging artifacts.

[1] Department of Bio and Brain Engineering, Korea Advanced Institute of Science and Technology, Daejeon, South Korea. [2] Department of Physiology and Biomedical Sciences, Seoul National University College of Medicine, Seoul, South Korea. [3] Division of Integrative Biosciences and Biotechnology, Pohang University of Science and Technology, Pohang, South Korea. [4] Department of Mechanical Engineering, Pohang University of Science and Technology, Pohang, South Korea. [5] Neuroscience Research Institute, Seoul National University College of Medicine, Seoul, South Korea. [6] Kim Jaechul Graduate School of AI, Korea Advanced Institute of Science and Technology, Daejeon, South Korea. ✉email: jong.ye@kaist.ac.kr

Three-dimensional (3D) fluorescence imaging reveals important structural information about a biological sample that is typically unobtainable from a two-dimensional (2D) image. Recent advancements in tissue-clearing methods[1–5] and light-sheet fluorescence microscopy (LSFM)[6–9] have enabled streamlined 3D visualization of biological tissue at an unprecedented scale and speed, sometimes even in finer details. Nonetheless, the spatial resolution in 3D fluorescence microscopy is still far from perfection; an isotropic resolution remains difficult to achieve.

Anisotropy in fluorescence microscopy typically refers to more blurriness in the axial imaging plane. Such spatial imbalance in resolutions can be attributed to many factors, including diffraction of light, axial undersampling, and the degree of aberration correction. Even for super-resolution microscopy[10], which in essence surpasses the light diffraction limits, such as 3D-structural illumination microscopy (3D-SIM)[11,12] or stimulated emission depletion (STED) microscopy[13], matching the axial resolution to the lateral resolution remains a challenge[14]. While LSFM, where the fluorescence-excitation path does not necessarily align with the detection path, provides a substantial enhancement to the axial resolution[9], a truly isotropic point spread function (PSF) is difficult to achieve for most contemporary light-sheet microscopy techniques, and the axial resolution is usually 2 or 3 times worse than the lateral resolution[15–17].

In the recent years of image restoration in fluorescence microscopy, deep learning emerged as an alternative, data-driven approach to replace the classical deconvolution algorithms. Deep learning has its advantage in capturing the statistical complexity of an image mapping and enabling end-to-end image transformation without painstakingly fine-tuning the parameters by hand. Some examples include improving the resolution across different imaging modalities and numerical aperture sizes[18], towards isotropy[19,20], or less noise[19]. While these methods provide some level of flexibility in the operation of microscopy, these deep-learning-based methods must assume some knowledge of a target data domain for the network training. For example, for isotropic reconstruction, Weigert et al[19,20]. used a supervised-learning strategy of pairing high-resolution lateral images with low-resolution axial images that were blurred with an explicit PSF model. Zhang et al[21]. implemented a GAN-based super-resolution technique with an image degradation model taken from the microscope. In both cases, the image degradation process is not dynamically learnable, and such an assumption of a fixed image degradation process requires the success of image restoration to rely on the accuracy of priors and adds another layer of operation to microscopists. Moreover, if the initial assumption of the image degradation is not correct, the performance in a real-world data set may be limited. Especially for high-throughput volumetric fluorescence imaging, the imaging conditions are often subject to fluctuation, and the visual characteristics of samples are considered diverse. Consequently, uniform assumption of prior information throughout a large-scale volume image could result in over-fitting of the trained model and exacerbate the performance and the reliability of image restoration.

In light of this challenge, the recent approach of unsupervised learning using cycle-consistent generative adversarial network (cycleGAN)[22] is a promising direction for narrowing down the solution space for ill-posed inverse problems in optics[23,24]. Specifically, it is advantageous in practice as it does not require matched data pairs for training. When formulated as an optimal transport problem between two probability distributions[23,25], unsupervised learning-based deconvolution microscopy can successfully transport the distributions of blurred microscopy images to high-resolution microscopy images by estimating the blurring

PSF and deconvolving with it[24]. Accordingly, it is less prone to generating artificial features compared to GAN[26] as theoretically analyzed in a prior work[23]. Moreover, if the structure of the PSF is partially or completely known, one of the generators could be replaced by a simple operation, which significantly reduces the complexity of the cycleGAN and makes the training more stable[24]. Nonetheless, one of the remaining technical issues is the difficulty of obtaining additional volumes of high-resolution microscopy images under similar experimental conditions, such as noise profiles and illumination conditions, so that they can be used as an unmatched target distribution for the optimal transport. In particular, obtaining such a reference training data set at a 3D isotropic resolution remains challenging in practice.

To address this problem, here, we present an unsupervised deep learning framework that blindly enhances the axial resolution in volumetric fluorescence microscopy, given a single 3D input image. The network can be trained with one image stack that has an anisotropic spatial resolution without requiring high-resolution isotropic 3D reference volumes. Thereby, the need to acquire additional training data sets under a similar experimental condition is completely avoided. Our framework takes advantage of forming abstract representations of objects that are imaged coherently in lateral and axial views: for example, 2D snapshots of neurons to reconstruct a generalized 3D neuron appearance. Then, our unsupervised learning scheme uses the abstract representations to decouple only the resolution-relevant information from the images, as well as undersampled or blurred details in axial images. This strategy translates to an advantage in applications for large-scale volume images, such as the entire cortical region of a brain. Figure 1a illustrates our approach. We demonstrated the success of the framework in simulation, confocal fluorescence microscopy (CFM), and open-top light-sheet microscopy (OT-LSM). In the CFM experiment, we addressed anisotropy that is mainly driven by light diffraction and axial under-sampling. We compared the results to a 3D image that was separately imaged at a perpendicular angle. In the OT-LSM experiment, our goal was to test whether our method can address anisotropy that is governed by a mixture of multiple image degrading factors, many of which are not simply modeled with a PSF convolution: e.g., motion artifacts from sample vibration by the stage drift. In all the cases, our reference-free deep-learning-based super-resolution approach was effective at improving the axial resolution, while preserving the information in the lateral plane and also restoring the suppressed microstructures.

## Results

**CyleGAN architecture**. The overall architecture of the framework was inspired by the optimal transport-driven cycle-consistent generative adversarial networks (OT-cycleGAN)[23]. Fig. 1b illustrates the learning scheme of the framework. We employ two 3D generative networks ($G$ and $F$ in Fig. 1b) that learn to generate an isotropic 3D image from an anisotropic 3D image (the forward or super-resolving path) and vice versa (the backward or blurring path), respectively. To curb the generative process of these networks, we employ two groups of 2D discriminative networks ($D_X$ and $D_Y$ in Fig. 1b). Our key innovation comes from an effective orchestration of the networks' learning based on how the discriminative networks sample during the learning phase. In the forward path, the discriminative networks of $D_X$ compare 2D axial projection images from the generated 3D image to 2D lateral images from the real 3D image, while preserving the lateral image information. Projection images from the generated volume are obtained as maximum intensity projections (MIP) with a randomized depth within a pre-determined range and are designed to emulate the lateral visual information projected from the adjacent

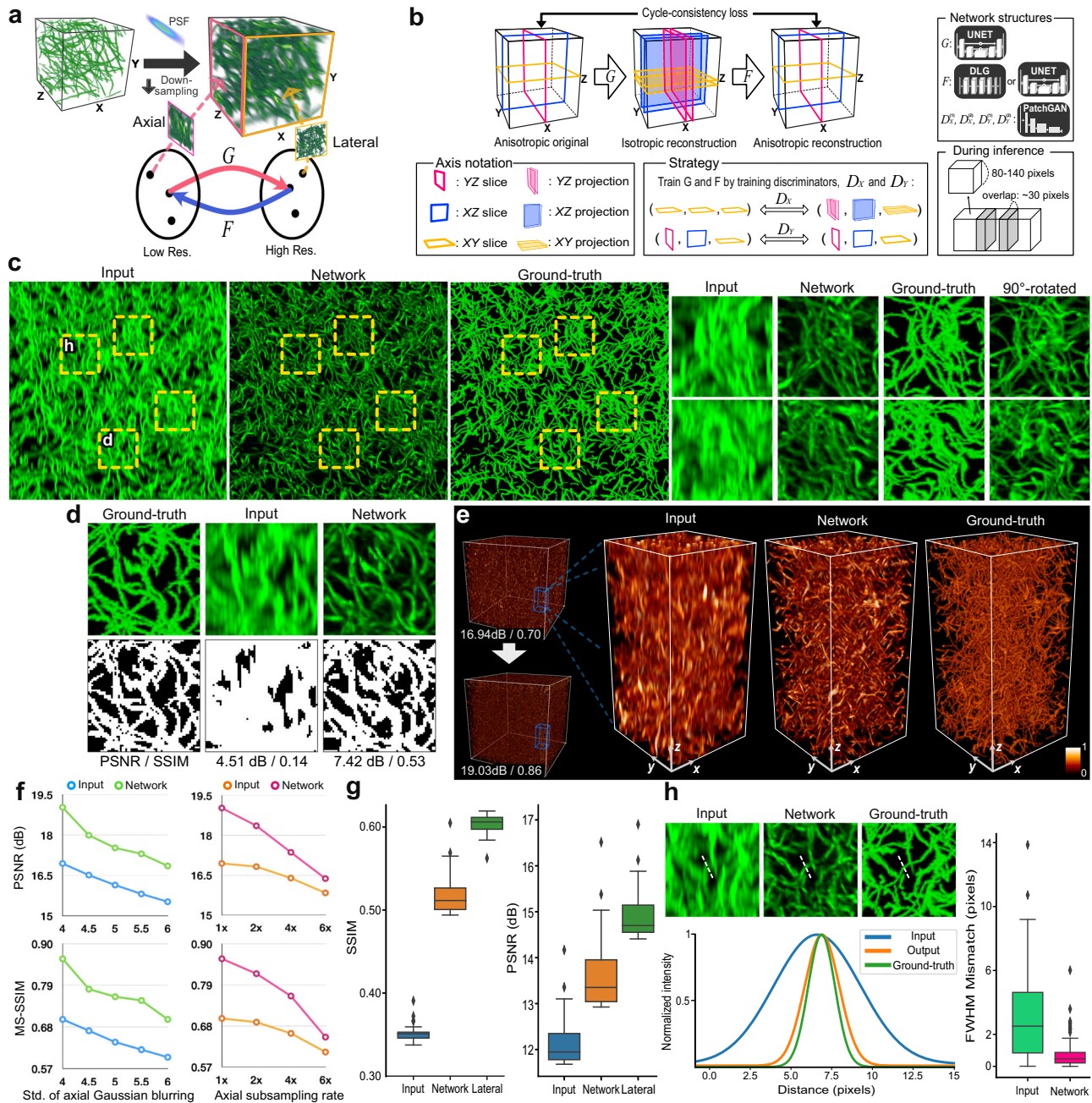

slices. This pairing of samples to train the discriminative networks encourages the 3D generative network $G$ to enhance only the axial resolution in the 3D volume output. On the other hand, the discriminative networks of $D_Y$ in the backward path compare 2D images from the reconstructed 3D image to 2D images from the real 3D image in each corresponding orthogonal plane; the 3D generative network $F$ learns to revert the image restoration process. The cycle-consistency loss stabilizes the learning process and guides $G$ and $F$ to being mutually inverse. By achieving the balance of loss convergence in the form of a mini-max game[26] by this ensemble of the discriminative and generative networks, the network $G$ is trained to learn the transformation from the original anisotropic resolution to the desired isotropic resolution.

**Simulation studies**. We initially simulated anisotropy in a 3D synthetic image and tested our framework for the task of blind deconvolution. The synthetic volume, of $900 \times 900 \times 900$ voxels, contains 10,000 tubular objects that were randomly placed and deformed by a 3D elastic grid-based deformation field. This simulation model allows the tubes to intertwine with each other in a non-linear manner and is ideal for simulating biological samples with complex mesh tubular structures, such as microtubules or neuronal networks. The image was then convolved with a Gaussian kernel that blurs only axially with a standard deviation of 4. Supplementary Fig. 1 visualizes this generation process. The networks were trained using one image sample, and, during training and inference, we used mini-batches with sub-regions of $120^3-144^3$ voxels. After inference, the sub-regions were stacked back to the original image space (Fig. 1b).

Figure 1c–h shows the results of blind deconvolution by the proposed method. After the blind deconvolution, the network output resolved the intricate entangling of the tubes, which was previously masked by the axial blurring, as visually illustrated in

**Fig. 1 Framework schematics and simulation studies. a** In fluorescence microscopy, 3D imaging is often subject to anisotropy that arises from light diffraction and under-sampling in the scanning direction. Our approach for single-sample super-resolution is to learn two-way transformations, G and F, between the high-resolution manifold and the low-resolution manifold, by sampling from lateral and axial slices or lateral and axial MIPs. **b** Schematic of the framework. The generative networks, G and F, learn the data-specific transformation mapping between low-resolution images and high-resolution images by learning to super-resolve axial images and revert the process, respectively. G and F use 3D convolution layers, and the discriminative networks, $D_X$ and $D_Y$, drive the learning process for G and F and use 2D convolution layers. Inference on a large-scale volume is carried out iteratively on sub-volumes with overlapping neighboring blocks. **c** Blind deconvolution results by our method, with zoomed-in ROIs (shown as yellow-dotted boxes) additionally for (**d**) and (**h**). **d** Assessment of reconstruction accuracy in 2D. The signals were segmented from the background using Otsu's method, shown with PSNR and SSIM for quantification. **e** 3D visualization of large-scale inference. A random region of $700^3$ voxels was selected as a test volume to calculate PSNR and MS-SSIM. The color bar represents the signal intensity normalized between 0 and 1. **f** Performance comparison using PSNR and MS-SSIM on the test volume under different imaging conditions: axial blurring and z-axis undersampling. The undersampling is done after the axial Gaussian blurring with a standard deviation of 4. **g** Performance comparison using SSIM and PSNR of MIP images. Cross sections ($n = 47$ non-overlapping independent samples) were taken with 15 slice depths to generate MIP images. "Lateral" refers to lateral sections of the volume, which was rotated perpendicularly before the blurring to provide a high-resolution reference. **h** Resolution improvement by FWHM mismatch with the ground truth, with cross-sectional intensity profiles from marked lines in zoomed-in ROIs from **c**. We measured the FWHM mismatches of tubular objects ($n = 317$ non-overlapping independent samples) with respect to the ground truth. The reconstruction by the network output exhibits a noticeable improvement in reconstruction accuracy. To calculate FWHMs, the intensity profiles were fitted into Gaussian functions. For the box plots plotted in (**g**) and (**h**) panels, the box shows the inter-quartile range (IQR) between the first quartile (Q1) and the third quartile (Q3) of the dataset, with the central mark (horizontal line) showing the median and the whiskers indicating the minimum (Q1-1.5*IQR) and the maximum (Q3+1.5*IQR). Outliers are represented by diamond-shaped markers beyond the whiskers.

the regions of interest (ROIs) in Fig. 1c, e. To quantify the resolution improvement on a large scale, we randomly selected a region of $700 \times 700 \times 700$ voxels and calculated the peak signal-to-noise ratio (PSNR) and multi-scale structural similarity index measure (MS-SSIM)[27] as metrics. We noticed that the PSNR and MS-SSIM metrics were higher in the 3D network output by approximately 2 dB and 0.16, respectively (Fig. 1e): input PNSR and MS-SSIM were 16.94 dB and 0.70, and output PSNR and MS-SSIM were 19.03 dB and 0.86. To evaluate the effect of image degradation on the network's performance, we trained and tested for different imaging conditions: the level of axial blurring and axial under-sampling. The level of axial blurring was controlled by the standard deviation of the Gaussian kernel, and the axial under-sampling was done in addition to the Gaussian blurring with a standard deviation of 4 and mimics the under-sampling process in fluorescence microscopy by taking slices with intervals: e.g., 4× means taking a slice every four slices in the z-axis. After subsampling, the images were resized back to the original dimensions by bi-linear interpolation. We noticed that throughout different levels of degradation, the network output was consistently higher in performance (Fig. 1f). As the goal of the learning was to enhance axial volumes using MIPs of the generated volumes, we also measured PSNR and structural similarity index measure (SSIM) of 47 $700 \times 700$-pixel axial MIP images and compared the results with corresponding lateral MIP images of the reference volume, which was imaged at a perpendicular angle, as the lateral resolution reference. The metrics of the network output were more closely approximated to those of the lateral MIP images (Fig. 1g).

In order to assess the reconstruction capability, we randomly selected 317 tubular objects and calculated the FWHM (full width at half maximum) mismatches of the input and output image with respect to the ground truth (Fig. 1h). The mean FWHM mismatch of the network output was approximately five times lower with a mismatch of 0.61 pixels compared to the 2.95-pixel mismatch in the input. The network output also showed a lower standard deviation of 0.60 pixels compared to 2.41 pixels in the input. As an alternative method, we also segmented signals from the background by using various histogram-based binarization methods: Otsu's method[28], Iterative Self-Organizing Data Analysis Technique[29] (ISODATA), and mean-thresholding[30]. Fig. 1d shows an example of segmentation by Otsu's method. For all the segmentation methods, the PSNR and SSIM metrics were consistently higher for the network output than the input

(Supplementary Fig. 2). To further assess the deconvolution accuracy of the output image, we also performed Fourier Spectrum analysis before and after deconvolution to better visualize the restoration of the high-frequency information and showed that in comparison to the input, the frequency information of the output is approximated more closely to that of the ground truth and its lateral counterpart (Supplementary Fig. 3). Furthermore, we noticed that our framework can be seen as interpretable during the training. We could approximate the blurring PSF model by calculating the impulse response through the generative network in the backward path, which was initially modeled as a linear blur kernel that emulates the axial blurring process (Supplementary Fig. 4).

**Resolution enhancement in confocal fluorescence microscopy.** We demonstrated the resolution improvement in the axial plane by imaging a cortical region of a Thy 1-eYFP mouse brain using CFM. The sample was tissue-cleared and was imaged in 3D using optical sectioning. The optical sectioning in CFM generated a stark contrast between the lateral resolution and axial resolution, with an estimated lateral resolution of 1.24 μm with a z-depth of 3 μm interval. The image volume, whose physical size spans approximately $1270 \times 930 \times 800$ μm³, was re-sampled for reconstruction isotropically to a voxel size of 1 μm using bilinear interpolation. In order to provide a reference that confirms the authenticity of the resolution improvement, we additionally imaged the sample after physically rotating it by 90 degrees, so that its high-resolution lateral XY plane would match the axial XZ plane of the original volume, while sharing the axial YZ plane. The reference volume was then registered on a cell-to-cell level to the input image space using the BigWarp Plugin[31]. While the separately acquired reference is far from a perfect ground truth image because of its independent imaging condition and potential registration error, it still provides the best available reference as to whether the details reconstructed by the framework match the real physical measurements.

In our test on the CFM volume, the trained network restored the previously highly anisotropic resolution to a near-perfect isotropic resolution. We illustrated the resolution improvement by comparing the distance on a resolved axial image and the corresponding distance in the lateral image for a structure that is symmetrical between the lateral and axial plane. One example, shown in Fig. 2a, is a basal dendrite, which is primarily cylindrical. In this example, the difference was nanoscale

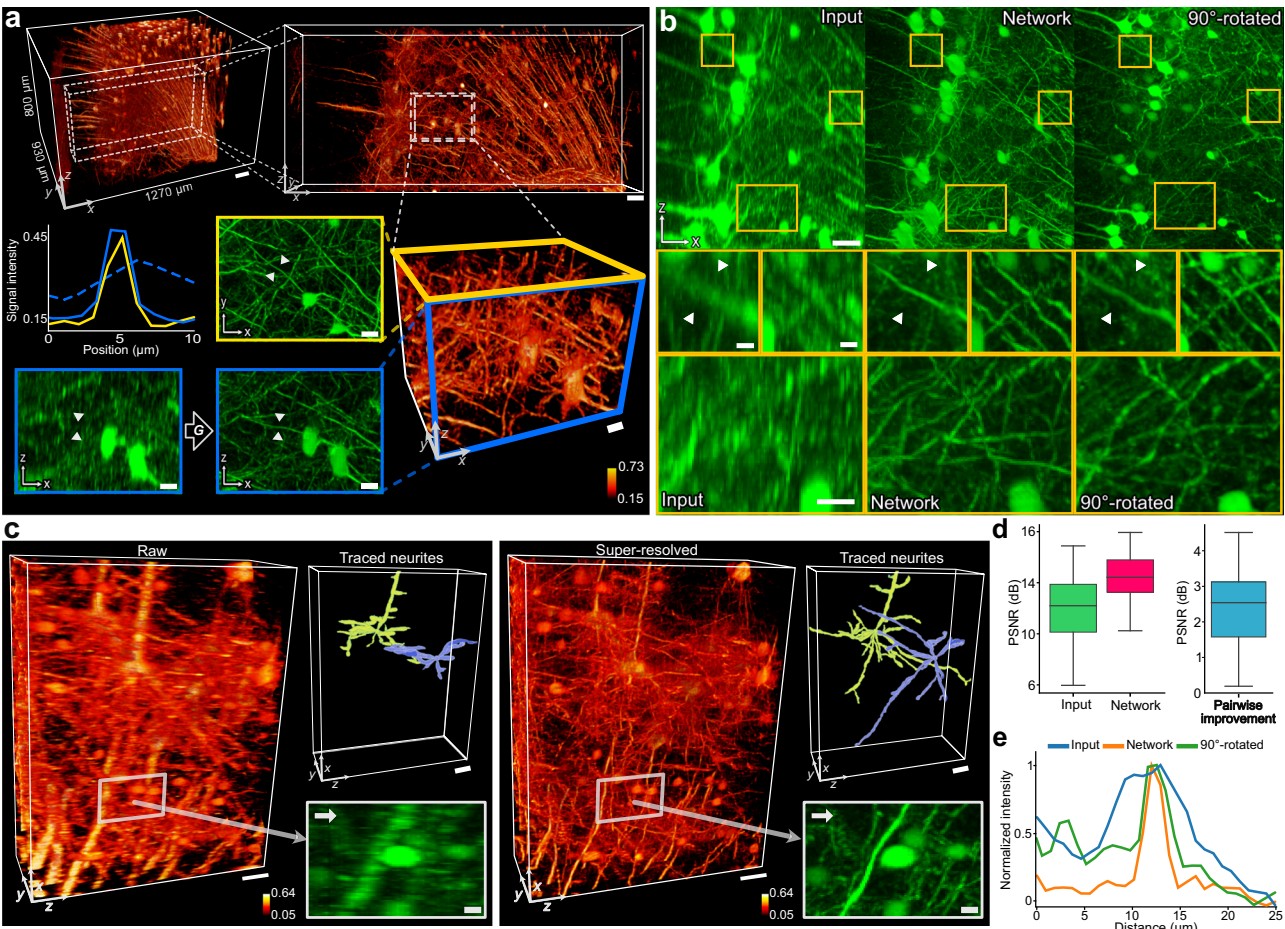

**Fig. 2 Deep learning enables single-volume super-resolution for volumetric CFM.** Cortical regions of a Thy 1-eYFP mouse brain were imaged. **a** Near isotropic resolution was achieved by the proposed method in a CFM volume. Axial images were blindly enhanced by the generative neural network, which is trained and tested on a single CFM volume that spans 1-2 gigabytes in memory. The resolution improvement in the axial planes was global and consistent throughout the image space (also see Supplementary Fig. 6). The converging cross-sectional intensity profiles of a cylindrical dendrite in the *XY* plane (yellow line) and *XZ* plane (blue line) indicate a near-perfect isotropic resolution in comparison to the input in *XZ* (dotted blue line), with *XY* FWHM of 1.71 µm and *XZ* FWHM of 1.76 µm, greatly reduced from input *XZ* FWHM of 6.04 µm. Scale bars: 100 µm, 50 µm, 20 µm (2D), and 20 µm (3D) in the progressively zooming order. **b** Image restoration results ("Network") show the upper cortical regions of the mouse cortex, as MIP images of 150-µm thickness. The results were compared to the original axial imaging ("Input") and reference lateral imaging ("90°-rotated"). Zoomed-in ROIs are marked as yellow boxes. Suppressed or blurred details were recovered in the network output images and matched the lateral imaging. Scale bars: 50 µm (top ROIs), 10 µm (middle ROIs), 20 µm (bottom ROIs). **c** 3D reconstruction of pyramidal neurons in the upper cortical layer before and after restoration, with neuronal tracings. Resolution improvement in the axial plane allows a more precise and detailed reconstruction of 3D neuronal morphology. We verified the additional neuronal tracings by substantiating in the corresponding locations from the lateral imaging. The under-sampling and the *Z*-blurring made it more difficult to trace neurites that run perpendicularly to the scanning direction (arrows in 2D ROIs), whereas more accurate tracing was possible from the network output. Scale bars: 50 µm (3D) and 10 µm (2D ROI). **d** PSNR distribution of MIP images as a distance metric to the reference image, with pairwise improvements. MIP images ($n = 31$ independent images) were from $140 \times 140 \ \mu m^2$ cross sections with depths of 150 µm. For the box plots, the box shows the IQR between Q1 and Q3 of the dataset, with the central mark showing the median and the whiskers indicating the minimum (Q1-1.5*IQR) and the maximum (Q3+1.5*IQR). **e** Cross-sectional intensity profiles from marked lines in zoomed-in ROIs from (**b**). The 90°-rotated line is registered to the input. In (**a**) and (**c**) panels, the color bars represent the signal intensity normalized between 0 and 1.

(~0.05 µm). As the textural differences between the lateral images and the super-resolved axial images were imperceptible to human eyes, we performed Fourier Spectrum analysis before and after restoration and showed that the frequency information of the output was restored to match that of the lateral imaging (Supplementary Fig. 5).

We examined the anatomical accuracy of the resolved details by comparing them to the reference image (labeled as 90°-rotated in Fig. 2b, c), which provides a high-resolution match to the original axial image on a micrometer scale. We noticed that the network was successful not only in translating the axial image texture to the high-resolution image domain, but also in recovering previously suppressed details, which were verified by the reference imaging. In accordance with the reference image, the network output accurately enhanced anatomical features of the nervous tissue, consistently throughout the image space (Fig. 2a and Supplementary Fig. 6). As shown in Fig. 2b, c, the network allowed for more advanced cytoarchitectonic investigation of the cortical region, as the network managed adaptive recovery of important anatomical features that vary in morphology, density and connectivity across the cortical region. For example, in the upper cortical layers, the previously blurred apical dendrites of pyramidal neurons were resolved (middle-left ROIs in Fig. 2b). The network output also revealed previously unseen

cortical micro-circuitry by pyramidal neurons and interneurons (bottom ROIs in Fig. 2b). The cross-sectional intensity profile (Fig. 2e) illustrates such recovery of suppressed details that were previously blurred in the axial imaging. We noticed that while the network improved the axial resolution, it introduced no discernible distortions or artifacts to the lateral plane.

Such recovery of suppressed details translated to noticeable improvement in reconstructing the neuronal morphologies from the enhanced output volume. We traced two pyramidal neurons by using NeuroGPS-tree[32], the current standard neuronal tracing method for instance segmentation, which was followed by human correction. The tracing was performed blindly without knowledge of other image counterparts. Figure 2c shows the results. In visual comparison, the tracings from the network output were not limited in any direction, whereas in the input image, the NeuroGPS-tree failed to trace neurites that were discontinued by the z-axis under-sampling (2D ROI in Fig. 2c). Comparisons of neuronal tracings from the input, network, and reference imaging are shown in Supplementary Fig. 7. To quantify the anatomical accuracy of the reconstructed details by the network, we substantiated the reconstructions by the network via slice-by-slice verification of the output tracings on images from the lateral imaging. The verification was performed on the level of examining the branching of reconstructed neurites by deletion of false positives. For example, traced image regions with non-matching neurites were set to zeros, and those with matching neurites were set to ones in a verified binary image. The tracing results and the verification process are further described in Supplementary Movies 1 and 2. Then, we measured the biological precision of the tracings by calculating the ratio between the original tracings and the verified tracings. The precisions of the neuron reconstruction from the two test neurons were 98.31% and 98.26%.

To quantify the axial resolution enhancement on a signal level, we identified 31 non-overlapping ROIs each of $140 \times 140 \, \mu m^2$ in the input axial images and the reference lateral images, where identical neuronal structures were distinguishable and detected similarly in visuals by fluorescence emission. Then, we measured and compared the peak signal-to-noise ratio (PSNR) distance of the input ROIs and the network output ROIs to the corresponding reference ROIs. The network introduced a mean PSNR improvement of 2.42 dB per pair of an input ROI versus an output ROI (Fig. 2d). This analysis suggests that the textural details recovered by the network include anatomically accurate features that were more discernible in the lateral imaging. We noticed that their metric differences were not entirely reflective of the perceptual accuracy of recovered details and were attributed to differences in fluorescence emission between the imaging sessions by imaging at a different angle (refer to Supplementary Fig. 8). We further tested for the generalization capability of the framework by training and testing for other biological cell or tissue types by imaging rat brain tissues with CFM: astrocytes labeled with fluorescent GFAP markers and blood vessels labeled with Lectin (refer to Supplementary Figs. 9 and 10). In our assessments, the framework showed biologically meaningful improvements in image quality and reconstruction.

**PSF-deconvolution capability.** Light-sheet microscopy (LSM) is a specialized microscopic technique for high-throughput cellular imaging of large tissue specimens including optically cleared tissues. To further explore the image restoration capability of our framework, we tested the deconvolution capability of an experimentally measured PSF on an open-top LSM (OT-LSM) system[33]. We imaged 0.5-μm fluorescent beads with OT-LSM, in the image stack with an overall physical size of

$360 \times 360 \times 160 \, \mu m^3$. The image was re-sampled for reconstruction isotropically to a voxel size of 0.5 μm using bilinear interpolation. The beads were spread arbitrarily, with some of the beads spaced closer to each other. The results by our framework are shown in Fig. 3. After the deconvolution, the 2D and 3D reconstruction of the network output indicated an almost isotropic resolution, resulting in a nearly spherical shape (Fig. 3a). This deconvolution effect was consistent across individual fluorescent beads. To quantify the performance of deconvolution, we calculated 2D FWHM values of more than 300 randomly selected bright spots and compared the lateral FWHM and the axial FWHM before versus after the image restoration. As shown in Fig. 3b, the FWHM distributions of the bright spots in the restored image show an almost identical match to those of the input in the lateral plane. The network output corrected the axial elongation of the PSF, with a mean axial FWHM of ~3.91 ± 0.28 μm being reduced to ~1.95 ± 0.12 μm, which is very close to the mean FWHM of ~1.98 ± 0.13 μm from the lateral input. The network introduced very little deviation in the lateral plane, with a mean FWHM mismatch of ~0.13 ± 0.06 μm.

**Resolution enhancement in light-sheet fluorescence microscopy.** Imaging a large-scale sample at a high resolution, such as imaging a whole mouse brain at a sub-micrometer resolution, may introduce an aggregate of unexpected image artifacts that are not noticeable at a lower resolution. In LSM microscopy, these artifacts are often by-products of a poorly calibrated or installed microscope. In particular, standard OT-LSM systems[34,35] require the excitation path and the imaging path to be perpendicular to each other and may introduce distortions to the image quality unevenly between the *XZ* plane and *YZ* plane, although this anisotropy can be relaxed by tightly focused excitation[33]. To blindly test our framework on a combination of unknown image degradation processes, we trained and tested on a OT-LSM system with multiple imaging artifacts, which include not only the blurring artifacts by spherical aberration that is caused by the refractive index mismatch between air and immersion medium, but other artifacts that span non-uniformly across the image space: for example, image doubling artifacts by missed synchronization between the sweeping of the excitation laser and the rolling shutter of the detection sensor, or motion blur artifacts from physical sample drifts by the motorized stage. We tested our framework to address these issues blindly, from a single session imaging with no priors from any fiduciary markers or other conventional methods to prevent these artifacts. As the anisotropy problem for this OT-LSM system requires the network to learn two distinct image transformations in each orthogonal plane and is inconsistent across the image space, we implemented a variation of the framework that employs separate discriminators for the *YZ* plane and *XZ* plane and replaces the projection sampling with slice sampling for the discriminative networks.

We imaged the cortical region of a tissue-cleared mouse brain labeled with Thy 1-eYFP using OT-LSM, which has a physical size of ~$930 \times 930 \times 8600 \, \mu m^3$. The image was re-sampled for reconstruction isotropically to a voxel size of 0.5 μm using bilinear interpolation. The microscopy system is estimated to have an image resolution of ~0.5 μm laterally and ~4.6 μm axially, with a z-depth scanning interval of 1 μm, and generated multiple imaging artifacts as listed above. For the schematics of the OT-LSM system, refer to Supplementary Fig. 11. As shown in Fig. 4a, we noticed that the image quality degradation in axial images by the OT-LSM system was non-identical between the *XZ* and *YZ* plane, as *XZ* and *YZ* images were affected in different degrees by light propagation that aligns with the *Y*-axis and vibration from the lateral movement of the tissue-cleared sample while scanning.

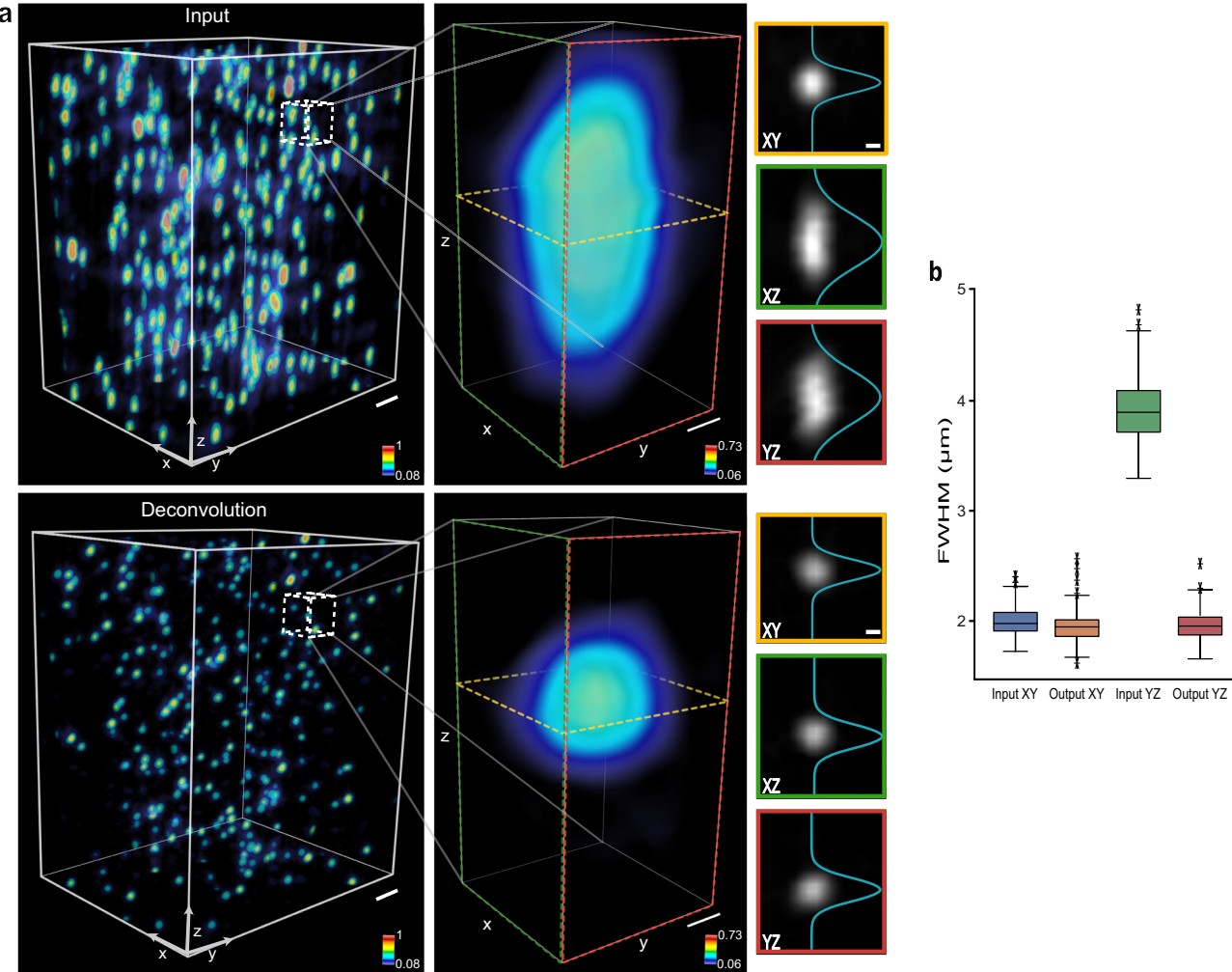

**Fig. 3 PSF Deconvolution by the framework.** The 0.5-μm fluorescent beads were imaged to model the PSF of the OT-LSM system experimentally. **a** An example of PSF deconvolution visualized in 3D and 2D. The intensity profiles were fit into Gaussian functions. Axial elongation is a common issue in fluorescence microscopy. Our framework resolves it to the originally spherical fluorescent bead. Slice images were visualized on the signal range of the input image. The color bars represent the signal intensity normalized between 0 and 1. Scale bars: 10 μm (left), 1 μm (middle and right). **b** FWHMs of experimentally measured PSFs in the lateral and axial planes before and after PSF deconvolution. We extracted bright spots from the same locations before applying the method ($n = 300$ spots for the $XY$ plane and 305 spots for the $YZ$ plane from non-overlapping distinct regions). Each spot was fit into a 2D Gaussian function, where FWHM was calculated. The PSFs in the axial plane were deconvolved to the almost identical resolution as those in the lateral plane. For the box plot, the box shows the IQR between Q1 and Q3 of the dataset, with the central mark showing the median and the whiskers indicating the minimum (Q1-1.5*IQR) and the maximum (Q3+1.5*IQR). Outliers are represented by asterisk-shaped markers beyond the whiskers.

As shown in Fig. 4c, the network output showed an evenly enhanced resolution between the $XZ$ and $YZ$ planes, while enhancing the contrast between signals and the background. This improvement enabled a more detailed reconstruction of 3D neuronal morphologies (Fig. 4b). For a visual comparison of the restored details, we de-convolved the image volume using the Richardson-Lucy (RL) deconvolution algorithm[36,37] based on the PSF model that was experimentally acquired (Fig. 3). The RL deconvolution was performed with Fiji-Plugin DeconvolutionLab[38] with 10 iterations. In the RL-deconvolution image, we found matching details that were previously suppressed by spherical aberration in the input image (Fig. 4c). Resolution improvements in both axial planes had imperceptible differences in texture and accuracy. However, as the image degradation was irregular across the image space, the RL-deconvolution image failed to address the additional imaging artifacts that were not modeled by the given PSF. In contrast, the network corrected many of these imaging artifacts. For example, the image doubling artifacts from the asynchrony between the excitation and detection were visibly reduced ($XZ$-plane

images of Fig. 4b and $XZ$-plane ROIs in Fig. 4c). The horizontal ripple artifacts caused by the stage drift were also corrected, ($YZ$-plane ROIs in Fig. 4c and Supplementary Fig. 12). We noticed that the artifact correction was consistent throughout the image space.

For quantitative assessment of the framework in the LSM system, we generated near-isotropic ground-truth images by calibrating the microscope[33], to a lateral resolution of $\sim 0.5\ \mu m$ and axial resolution of $\sim 1.9\ \mu m$, and greatly reducing most of the previous imaging artifacts. Then, we simulated a depth-wise PSF blurring process by applying an axial Gaussian kernel with a standard deviation of 10. We trained and tested on an image volume of $\sim 490 \times 130 \times 150\ \mu m^3$. In our tests with the original model, the network successfully recovered most of the blurred axial blurred information. Supplementary Fig. 13 shows the results of this experiment. In comparison to the RL-deconvolution image, the network output images exhibit recovery of fine details (zoomed-in ROIs in Supplementary Fig. 13a). Furthermore, we noticed that in comparison to the ground truth, the network also corrected the horizontal stripe artifacts, which were caused by the

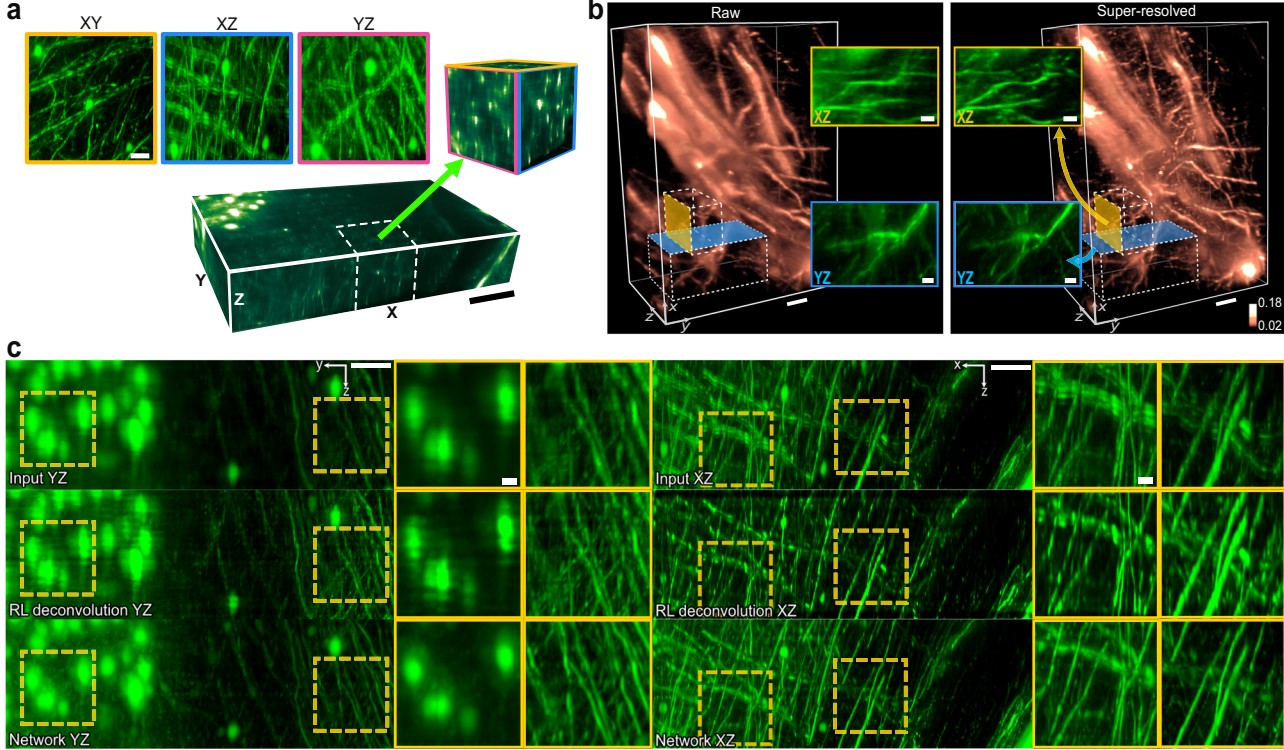

**Fig. 4 Image restoration by the proposed framework in large-throughput imaging using OT-LSM. a** Comparison of image qualities between the orthogonal image planes in an uncalibrated OT-LSM, using MIP images of a selected 3D ROI. Scale bar: 100 μm and 25 μm (zoomed-in ROIs). **b** 3D reconstructions of somata and basal dendrites of pyramidal neurons, with axial MIP images of selected 3D ROIs. The network corrected the doubling effect in the *XZ* plane and also enhanced the contrast between the signals and the background. The resolution enhancement was consistent across the *XZ* plane and *YZ* plane. The input and output images were visualized on the same intensity spectrum. The color bar represents the signal intensity normalized between 0 and 1. Scale bars: 25 μm. **c** *YZ* and *XZ* MIP images from the input, the deconvolution result by the Richardson-Lucy algorithm, and the network output. Deconvolution and artifact correction in the network output was consistent throughout 8600 slice images of the OT-LSM volume. The MIP images are in depths of 150 μm. Experiments were repeated with six independent image volumes, achieving similar results. Scale bars: 50 μm and 10 μm (zoomed-in ROIs).

axial under-sampling and illustrated in the ground-truth ROIs in Supplementary Fig. 13a. To quantify the axial resolution improvement, we divided the test volume into eight non-overlapping regions and calculated the image quality metrics on their respective MIPs with 17.5 μm depths. PSNR and MS-SSIM were used as the reference-aware metrics, and BRISQUE[39] was used as the no-reference image quality metric to assess the perceptual naturalness of the output image, in comparison to the input and the ground truth. We noticed improvements in all the metrics (Supplementary Fig. 13b). The BRISQUE metric suggests that the output images were perceptually more natural than the blurred input, with little difference from the ground truth. We also noticed an overall PSNR improvement of 1.98 dB in the 3D output volume.

So far, we have demonstrated the effectiveness of our method in simulation, CFM, and OT-LSM, which involve many dissimilarities from one another in image formation. Accordingly, we expect the framework to be widely applicable to other forms in the fluorescence microscopy spectrum, as the essential component of the learning does not rely on the conditions of an image formation process.

## Discussion

In this work, we developed a deep-learning-based super-resolution technique that enhances axial resolution of conventional fluorescence microscopy by learning from high-resolution lateral images. The strength of our framework comes from taking advantage of unsupervised learning from unmatched data pairs: it allows the learning of image transformation to be localized to a user-defined 3D unit space and thus to be decoupled from regional variations in image characteristics, such as aberrations or artifacts that arise naturally from a fluorescence imaging process. In our experiments with simulation, CFM, and OT-LSM, we showed that this feature translated to a distinct advantage for isotropic reconstruction of suppressed details in large-scale volumetric fluorescence microscopy, where the level of image degradation varies across the image space.

Deep-learning models based on a GAN architecture, which is of unsupervised learning, excel at generating very plausible high-level details and are known for tasks such as transferring art styles[22,40] or even tweaking high-level details[41]. However, when it comes to enhancing microscopy images in biology research, well-generated details are a double-edged sword. The high plausibility of the details makes it difficult to validate the authenticity of restored images, especially without referring to real biological evidence. In this paper, we provided both theoretical evidence from the simulation studies and experimental evidence from the orthogonal imaging and artificial blurring to substantiate the validity of such reconstructed high-frequency information. The results supported that our OT-cycleGAN network design addressed the issue of deviation from authenticity by strictly confining the solution space by the formulation of optimal transport and the physics-inspired design of the degradation model in the backward path, while harnessing this excellent generative power for reconstructing high-frequency information. Additionally, our ablation studies provide a deeper look into the essential components of our framework (Supplementary Note 1 and Supplementary Fig. 14).

In practice, our method is not to be mistaken for a one-size-fits-all solution to every anisotropy problem in microscopy. If necessary, users can include additional steps for visualization in post-processing. For example, some visual artifacts may emerge depending on how images are visualized: 2D versus 3D or the intensity range for visualization. Supplementary Fig. 15 displays an example of such cases. As the discriminative networks are trained with 2D MIP images, such visual artifacts may appear more discernible when visualized solely as 2D MIP images. However, these artifacts do not arise from false reconstructions, as shown in Supplementary Fig. 15; there is no spurious structure when fully visualized in 3D. In cases of the image volumes where the local contrast changes noticeably between sub-volumes during training iterations, while most artifacts are set within very low-intensity ranges and remain barely visible, they may seem more visible when visualized on a highly saturated intensity range. In such cases, users can normalize the output image locally according to the histogram of the input image, using histogram-matching[42]. Supplementary Fig. 16 shows our test with the image volume from the PSF deconvolution experiment; the post-processing alone increased the local signal-to-noise ratio (SNR).

Finally, the main benefit of our method lies in its ease of deployment. As shown in Supplementary Figs. 9 and 10, the framework is not necessarily limited by the type of imaged tissues or fluorescent labeling markers. Accordingly, our method should be applicable to a variety of imaging scenarios in volumetric fluorescence microscopy. It also greatly reduces the effort to be put into practice as training a network requires only a single 3D image stack, without a priori knowledge of the image formation process, registration of training data, or separate acquisition of target data. Some combination of those factors is generally considered necessary[43] for most conventional deep-learning-based super-resolution methods. For this reason, we expect our method to significantly lessen the pre-existing difficulty of applying super-resolution to volumetric microscopy data.

## Methods

**Simulation setup.** To simulate a randomized mesh structure with tubular objects, we first randomly selected 20,000 points in a 3D image space of $900 \times 900 \times 900$ voxels to draw 10,000 linear lines of two-pixel thickness. Then, we applied a 3D elastic grid-based deformation field with 70 grid locations with a sigma value of 3. The deformed volume was then normalized and treated as the ground-truth volume. To obtain a blurred volume, we applied a Z-blurring Gaussian kernel with a standard deviation of 4. Supplementary Fig. 1 visualizes this process.

**Sample preparation and image acquisition.** Tg(Thy1-eYFP)MJrs/J mice were identified by genotyping after heterozygous mutant mice were bred, and the mice were backcrossed onto the C57BL/6 WT background for 10 generations and then maintained at the same animal facility at the Korea Brain Research Institute (KBRI). Mice were housed in groups of 2–5 animals per cage with ad libitum access to standard chow and water in a 12/12 light/dark cycle with "lights-on" at 07:00, at an ambient temperature of 20-22 °C and humidity (about 55%) through a constant airflow. The well-being of the animals was monitored on a regular basis. All animal procedures followed the animal care guidelines approved by the Institutional Animal Care Use Committee (IACUC) of KBRI (IACUC-18-00018). In preparation for mouse brain slices, the mice were anesthetized by injection with a zoletil (30 mg/kg) and xylazine (10 mg/kg body weight) mixture. Mice were perfused with 20 ml of fresh cold phosphate buffered saline (PBS) and 20 ml of 4% paraformaldehyde (PFA) solution using a peristaltic pump and whole mouse brains were extracted and fixed in 4% PFA for 1–2 days at 4 °C. The fixed mouse brains were sliced coronally in 500 μm thickness. Then, the brain slices were incubated in a refractive index (RI) matching solution (C match, 1.46 RI, Crayon Technologies, Korea) at 37 °C for one hour for the optical clearing. The proposed method was applied to the images of the optically cleared tissue samples. For the sample preparation for Supplementary Figs. 9 and 10, refer to Supplementary Note 2.

For the CFM imaging, the optically cleared tissue specimens were mounted on a 35-mm coverslip bottom dish and were immersed in the RI matching solution during image acquisition using an upright confocal microscope (Nikon C2Si, Japan) with a Plan-Apochromat 10× lens (NA = 0.5, WD = 5.5 mm). The Z-stacks of optical sections were taken at 3 or 4 μm intervals.

For the OT-LSM imaging, we used a recently developed microscopy system[33], whose design is based on the water-prism open-top light-sheet microscopy[34,35]. To induce anisotropy, we excluded the tight focusing of the excitation light sheet across the imaging field of view. The system includes an ETL (EL-16- 40-TC-VIS-5D-M27, Optotune) as part of the illumination arm for the axial sweeping of the excitation light sheet and an sCMOS camera (ORCA-Flash4.0 V3 Digital CMOS camera, Hamamatsu) in the rolling shutter mode to collect the emission light from the sample. The system uses a 10× air objective lens (MY10X-803, NA 0.28, Mitutoyo) in both the illumination and imaging arms, pointing toward the sample surface at +45° and -45°. The custom liquid prism was filed with the RI matching solution for the normal light incidence onto the clearing solution. The excitation light source was either 488 nm or 532 nm CW lasers (Sapphire 488 LP-100, Coherent; LSR532NL-PS-II, JOLOOYO).

**Image pre-processing.** For the OT-LSM brain images, a median filter of a 2-pixel radius was applied to remove the salt-and-pepper noise that arises from fluorescence imaging. All images were normalized to scale affinely between 0 and 1 using percentile-based saturation with the bottom and top 0.25% for the synthetic images, 0.03% for the CFM images, and 3% for the OT-LSM images. In both OT-LSM experiments, since the OT-LSM system images a sample at 45°or −45°, we applied shearing in the YZ plane as an affine transformation to reconstruct the correct sample space. Image visualization was performed using the Fiji software[44] and Paraview[45].

**Neuron tracing and verification.** Two pyramidal neurons were chosen for the visibility of their connecting neurites. They were first automatically traced using the NeuroGPS-Tree software. The soma locations were manually selected using the V3D software[46]. The NeuroGPS-tree software used these soma locations for tracing, with the parameters of binary threshold and trace threshold, which were chosen empirically via manual observation and correction. After the initial tracing, the tracings were converted to a binary image volume and corrected manually based on its slice-by-slice comparison with the source image volume, setting the matching neurites to ones and the rest to zeros. In the verification step, the tracings were then translated to the corresponding locations in the registered reference volume, which was separately imaged after a 90-degree physical rotation of the sample. The verification was done slice-by-slice by verifying the tracings on the reference image slices (Supplementary Movies 1 and 2). For example, the image regions with non-matching neurites were set to zero, and the image regions with matching neurites were set to one. The precision was then calculated as below:

$$\text{precision} = \frac{\sum_{i=1}^{W}\sum_{j=1}^{H}\sum_{k=1}^{D} V(i,j,k)}{\sum_{i=1}^{W}\sum_{j=1}^{H}\sum_{k=1}^{D} C(i,j,k)} \quad (1)$$

Here, W, H, and D represent the width, height, and depth of the traced volume. V and C represent the binary verified volume after the referential verification and the binary corrected volume before the referential verification, respectively.

**Cycle-consistent generative adversarial network structure.** To derive our algorithm, we assume that the high-resolution target space $\mathcal{X}$ consists of imaginary 3D image volumes with an isotropic resolution according to a probability measure μ, while the input space $\mathcal{Y}$ consists of measured 3D volumes with an anisotropic resolution with a poorer axial resolution that follows a probability measure $\nu$. According to the optimal transport-driven cycleGAN[23], if we were to transform one image volume in $\mathcal{Y}$ to $\mathcal{X}$, we can solve this problem by transporting the probability distribution $\nu$ to μ and vice versa in terms of the statistical distance minimization in $\mathcal{X}$ and $\mathcal{Y}$ simultaneously[23], which can be implemented using a cycleGAN. In our implementation, it is the role of the discriminative networks to estimate such statistical distances and guide the generative networks to minimize the distances. Unfortunately, as $\mathcal{X}$ consists of imaginary isotropic high-resolution volumes, we cannot directly measure the statistical distance to $\mathcal{X}$ from the generated volumes. This technical difficulty can be resolved by the following observation: since isotropic resolution is assumed for every 3D volume $\boldsymbol{x} \in \mathcal{X}$, the XY, YZ, and XZ planes should have the same resolution as the lateral resolution of the input volume (i.e., the XY plane of the input volume $\boldsymbol{y} \in \mathcal{Y}$). Accordingly, we can measure the statistical distance to the imaginary volumes in $\mathcal{X}$ by defining the statistical distance as the sum of the statistical distances in the XY, YZ, and XZ planes using the following least square adversarial loss:[47]

$$\mathcal{L}_{\mathcal{Y}\to\mathcal{X}}\left(G, D_X\right) = \mathcal{L}_{\mathcal{Y}\to\mathcal{X}}\left(G, D_X^{(1)}\right) + \mathcal{L}_{\mathcal{Y}\to\mathcal{X}}\left(G, D_X^{(2)}\right) + \mathcal{L}_{\mathcal{Y}\to\mathcal{X}}\left(G, D_X^{(3)}\right) \quad (2)$$

where

$$\mathcal{L}_{\mathcal{Y}\to\mathcal{X}}\left(G, D_X^{(1)}\right) = \mathbb{E}_{\boldsymbol{y}\sim\nu}\left[\left(D_X^{(1)}(\boldsymbol{y}_{xy}) - 1\right)^2\right] + \mathbb{E}_{\boldsymbol{y}\sim\nu}\left[\left(D_X^{(1)}\left(\left[G(\boldsymbol{y})\right]_{xy_{proj}}\right)\right)^2\right] \quad (3)$$

$$\mathcal{L}_{\mathcal{Y}\to\mathcal{X}}\left(G, D_X^{(2)}\right) = \mathbb{E}_{\boldsymbol{y}\sim\nu}\left[\left(D_X^{(2)}(\boldsymbol{y}_{xy}) - 1\right)^2\right] + \mathbb{E}_{\boldsymbol{y}\sim\nu}\left[\left(D_X^{(2)}\left(\left[G(\boldsymbol{y})\right]_{xz_{proj}}\right)\right)^2\right] \quad (4)$$

$$\mathcal{L}_{\mathcal{Y}\to\mathcal{X}}\left(G, D_X^{(3)}\right) = \mathbb{E}_{\boldsymbol{y}\sim\nu}\left[\left(D_X^{(3)}(\boldsymbol{y}_{xy}) - 1\right)^2\right] + \mathbb{E}_{\boldsymbol{y}\sim\nu}\left[\left(D_X^{(3)}\left(\left[G(\boldsymbol{y})\right]_{yz_{proj}}\right)\right)^2\right] \quad (5)$$

where the subscripts xy, yz and xz refer to slice information on the xy, yz and xz plane, and the subscripts $xy_{proj}$, $yz_{proj}$ and $xz_{proj}$ refer to maximum intensity

projections on the *XY*, *YZ*, and *XZ* plane. The projection takes into account z-blurred projections on the lateral plane from adjacent slices. Here, $\mathbf{y}_{xy}$, a *XY* 2D slice image from the image volume **y**, is used as the *XY*, *YZ*, and *XZ* plane references from the imaginary isotropic volume distribution $\mathcal{X}$ and compared with the corresponding planes of the restored volume G(**y**).

On the other hand, the backward path discriminator group $D_Y$ is trained to minimize the following loss:

$$\mathcal{L}_{\mathcal{X}\rightarrow\mathcal{Y}}\left(F, D_Y\right) = \mathcal{L}_{\mathcal{X}\rightarrow\mathcal{Y}}\left(F, D_Y^{(1)}\right) + \mathcal{L}_{\mathcal{X}\rightarrow\mathcal{Y}}\left(F, D_Y^{(2)}\right) + \mathcal{L}_{\mathcal{X}\rightarrow\mathcal{Y}}\left(F, D_Y^{(3)}\right) \quad (6)$$

where

$$\mathcal{L}_{\mathcal{X}\rightarrow\mathcal{Y}}\left(F, D_Y^{(1)}\right) = \underset{\mathbf{y}\sim\nu}{\mathbb{E}}\left[\left(D_Y^{(1)}(\mathbf{y}_{xy}) - 1\right)^2\right] + \underset{\mathbf{x}\sim\mu}{\mathbb{E}}\left[\left(D_Y^{(1)}\left([F(\mathbf{x})]_{xy}\right)\right)^2\right] \quad (7)$$

$$\mathcal{L}_{\mathcal{X}\rightarrow\mathcal{Y}}\left(F, D_Y^{(2)}\right) = \underset{\mathbf{y}\sim\nu}{\mathbb{E}}\left[\left(D_Y^{(2)}(\mathbf{y}_{xz}) - 1\right)^2\right] + \underset{\mathbf{x}\sim\mu}{\mathbb{E}}\left[\left(D_Y^{(2)}\left([F(\mathbf{x})]_{xz}\right)\right)^2\right] \quad (8)$$

$$\mathcal{L}_{\mathcal{X}\rightarrow\mathcal{Y}}\left(F, D_Y^{(3)}\right) = \underset{\mathbf{y}\sim\nu}{\mathbb{E}}\left[\left(D_Y^{(3)}(\mathbf{y}_{yz}) - 1\right)^2\right] + \underset{\mathbf{x}\sim\mu}{\mathbb{E}}\left[\left(D_Y^{(3)}\left([F(\mathbf{x})]_{yz}\right)\right)^2\right] \quad (9)$$

so that *XY*, *YZ*, and *XZ* plane images of the blurred volume $F(\mathbf{x})$ follow the distribution of *XY*, *YZ*, and *XZ* plane images of the input volume $\mathbf{y} \in \mathcal{Y}$.

Note that *G* is a 3D generator that performs 3D deblurring or upsampling on an input image volume. During this 3D restoration process, there are chances that original *XY* slice images can be distorted. Therefore, the generator *G* should be trained to improve the resolution in both *XZ* and *YZ* slices, but also not to degrade the performance in the *XY* plane. This issue also concerns the generator *F* in the backward path. Therefore, we need discriminators for both axial sampling and lateral sampling during the 3D restoration step in the forward path and the 3D degradation step in the backward path. Furthermore, if there is no discrepancy in the image quality between the two axial planes, only one discriminative network can be used to learn from both axial planes (i.e., $D_X^{(2)} = D_X^{(3)}$ and $D_Y^{(2)} = D_Y^{(3)}$). For example, the simulation studies and the CFM experiment were examples of this case.

Then, the full objective for the neural network training is given by:

$$\mathcal{L}\left(G, F, D_X, D_Y\right) = \mathcal{L}_{\mathcal{X}\rightarrow\mathcal{Y}}\left(F, D_Y\right) + \mathcal{L}_{\mathcal{Y}\rightarrow\mathcal{X}}\left(G, D_X\right) + \lambda\mathcal{L}_{\lrcorner\dagger\llcorner}(G, F) \quad (10)$$

where $\mathcal{L}_{\lrcorner\dagger\llcorner}(G, F)$ refers to the cycle-consistency loss and is calculated as the sum of absolute differences, also known as the L1 loss, between $F(G(\mathbf{y}))$ and **y**. λ, as the weight of the cycle-consistency loss, is set at 10 in our experiments. The objective function of the cycle-consistency-preserving architecture aims to achieve the balance between the generative ability and the discriminative ability of the model as it transforms the image data to the estimated target domain as closely as possible, while also preserving the reversibility of the mappings between the domains. The generative versus discriminative balance is achieved by the convergence of the adversarial loss in both paths of the image transformation, as the generative networks learn to maximize the loss and the discriminative networks, as their adversary, learn to minimize the loss.

The resulting architecture consists of two deep-layered generative networks, each in the forward path and the backward path, and four or six discriminative networks, in two groups each in the forward path and the backward path. The schematic is illustrated in Fig. 1b, and the detailed descriptions of the network designs are shown in Supplementary Fig. 17 and Supplementary Notes 3 and 4.

The generative network *G* in the forward path is based on the 3D U-Net architecture[48], which consists of the downsampling path, the bottom layer, the upsampling path, and the output layer. On the other hand, the generative network *F* in the backward path is adjustable and replaceable based on how well the generative network can emulate the blurring or downsampling process. We empirically searched for an optimal choice between the 3D U-net architecture and the deep linear generator[49] without the downsampling step (refer to Supplementary Fig. 17b). We chose the deep linear generator as *F* for simulations, the CFM brain images, and the OT-LSM fluorescent bead images, and we chose the 3D U-Net as *F* for the OT-LSM brain images. The kernel sizes in the deep linear generator vary depending on the depths of the convolution layers (refer to Supplementary Fig. 17b).

**Algorithm implementation and training**. Before the training phase for the OT-LSM images, we diced the entire volume into sub-regions of $200^3 - 250^3$ voxels with overlapping adjacent regions of 20-50 voxels in depth. The number of sub-regions is ~3000 for the brain image data and ~580 for the fluorescent bead image data. Then, we randomly cropped a region for batch training per iteration and flipped it on a randomly chosen axis as a data augmentation technique. The crop size was $132 \times 132 \times 132$ voxels for the brain images and $100 \times 100 \times 100$ voxels for the fluorescent bead images.

While the axial resolution in OT-LSM differs between the *XZ* and *YZ* planes because of the illumination path aligning with the *YZ* axis, the axial resolution in the CFM imaging is consistent across the *XY* plane. For this reason, for the CFM images, we loaded the whole image volume (1-2 gigabytes) in memory and randomly rotated along the *Z*-axis as a data augmentation technique. Then, we randomly cropped a region and flipped on a randomly chosen axis per iteration. For this reason, the networks were trained with one whole image volume with its

training progress marked in iterations instead of epochs. The crop size is set as $144 \times 144 \times 144$ voxels. During the inference phase in all experiments, the crop size is set as $120 \times 120 \times 120$ voxels with overlapping regions of 30 voxels in depth, and we cropped out the borders by a depth of 20 voxels from each output sub-region to remove weak signals near the borders before assembling back to the original image space. In all experiments, the batch size per iteration is set as 1.

In the 3D U-net generative networks, all 3D convolution layers have the kernel size of 3, the stride of 1 with the padding size of 1, and all transposed convolution layers have the kernel size of 2, the stride of 2, and no padding. In the deep linear generative networks, the six convolution layers have the kernel sizes of [7,5,3,1,1,1] in turn with the stride of 1 and the padding sizes of [3,2,1,0,0,0]. In the discriminative networks, the convolution layers have the kernel size of 4, the stride of 2, and the padding size of 1. For the axial projection depth, we set it to a randomized depth at each iteration confined between 2 slices to 15 slices for the CFM imaging of the brain and between 2 slices to 10 slices for the simulation studies, the fluorescent bead imaging, the CFM imaging of astrocytes, and the synthetically blurred OT-LSM imaging.

In all experiments, all learning networks were initialized using Kaiming initialization[50] and optimized using the adaptive moment estimation (Adam) optimizer[51] with a starting learning rate of $1 \times 10^{-4}$. For the CFM images and the OT-LSM brain images, the training was carried out on a desktop computer with a GeForce RTX 3090 graphics card (Nvidia) and Intel(R) Core(TM) i7-8700K CPU @ 3.70 GHz. The training time of a model differed depending on the nature of the image and hyper-parameters such as the size of the ROI per training iteration. For example, training for a baseline model of the simulation was selected at the 11,000th iteration and took approximately 19 hours using a 16-bit image volume of $148^3$ voxels per iteration. The GPU memory consumption was at approximately 24 GB in this case. Inference took 3–5 min on a test volume of $700^3$ voxels. In our implementation, the U-Net model includes 7.077 million training parameters, the deep linear generator includes 0.647 million training parameters, and each discriminative network contains 2.763 million training parameters. The performance of the trained networks was measured using PSNR, SSIM, MS-SSIM[27], BRISQUE[39], and SNR. The definitions of PSNR, SSIM, and SNR in this study are described in Supplementary Notes 5, 6, and 7, respectively.

**Statistics and reproducibility**. Unless otherwise specified, all neural networks were trained once per set of hyper-parameters and input data. In terms of inference, all experiments were independently repeated at least three times per image volume, achieving similar results. Here, the proposed framework is reference-free and applied to large-scale images where biological features may vary across the image space. In all experiments, the entire image space was assessed and exhibited similar results.

**Reporting summary**. Further information on research design is available in the Nature Research Reporting Summary linked to this article.

## Data availability
Training and test data for the simulation, the CFM experiment, the OT-LSM experiment for PSF deconvolution, and the OT-LSM experiment with artificial blurring and test data for the OT-LSM experiment for artifact correction have been deposited in the Zenodo database under https://doi.org/10.5281/zenodo.6352948[52]. Training data for the OT-LSM experiment for artifact correction is available from the corresponding author upon reasonable request, due to size limitations. Source data are provided with this paper.

## Code availability
The code for network training and prediction (in Python/PyTorch) is publicly available at the Github repository[53].

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

## Acknowledgements

H.P. and J.C.Y. were supported by the National Research Foundation of Korea (Grant NRF-2020R1A2B5B03001980 and NRF-2017M3C7A1047904). The brain samples from transgenic mice were provided by Dr. Chang Man Ha of the Korea Brain Research Institute.

## Author contributions

H.P. conceived and implemented the research. J.C.Y. supervised the project in conception and discussion. M.N. generated the CFM imaging data. S.C. supervised the CFM imaging. B.K. and S.P. generated the OT-LSM imaging data. K.K. supervised the OT-LSM imaging. H.P. and J.C.Y. wrote the manuscript.

## Competing interests

The authors declare no competing interests.
