## [Peer Review File · Nature Communications]

Deep learning enables reference-free isotropic super-resolution for volumetric fluorescence microscopyREVIEWER COMMENTS

Reviewer #1 (Remarks to the Author):

The paper describes a deep learning (DL) based 3D deconvolution method for fluorescence microscopy that aims to restore isotropically resolved volumes from the typical axially underresolved input volumes without the need for isotropic ground truth (that is almost always impossible to obtain). Specifically they employ a 3D deblurring network G to estimate the isotropic volume and use a discriminatory (GAN) loss that enforces the obtained 2D slices along all axes to be comparable with the high resolved lateral slices of the original volume. Additionally, they use a blurring generator network F to enforce cycle consistency (cycleGAN). The method is demonstrated on a confocal and a lightsheet dataset of mouse brain tissue and a fluorescence bead dataset.

The paper is well written and the shown resolution improvements obtained on such large images are quite impressive. Furthermore, the idea to combine the different image resolution in sliced 2D views (lateral vs axial) together with the cycle consistency in a unsupervised deconvolution framework is innovative and addresses crucial practical obstacles that are present when dealing with deconvolution of large 3D microscopy datasets: i) the often missing availability of a good PSF estimate, and ii) the missing availability of isotropic ground truth (when using a supervised DL approach). So I think that this method could be of great practical importance and is certainly worthwhile to be communicated.

Although the authors show the merits of the method qualitatively on an impressive set of imaging experiments, I am however not fully convinced about the quantitative comparisons, which are currently quite limited. Furthermore, I would strongly argue that the authors make the code of their method available, as without that the practical impact of what appears to be a useful method would be rather limited. See below for detailed comments on these two major issues:

Major:

1) Quantification

Currently, the authors quantify their methods on extended structures (brain dataset) by lineplots (Fig1a,e) and the comparison against the lateral 2D slices of the 90 degree rotated reference (Fig1f). This is rather limited as it cannot describe the true 3D restoration improvement of the method. I understand that a truly isotropic 3D reference image would be impossible to obtain, but why did the authors not validate their method with a synthetically created dataset? This way, one could additionally compare against RL deconvolution in a truly quantitative way.

2) Software availability

Currently the authors provide no code that would allow other researchers to use this method on their own data. Specifically as the proposed method is unsupervised (and thus could be applied to already acquired 3D volumes) this would greatly increase the value of the paper.

Minor:

- It is worth to note, that the idea of using the lateral slices of a anisotropic microscopy volume as a target for the deconvolution of the poorer resolved axial slices is already used in the context of supervised DL deconvolution in the given reference [19]. This could be made clearer.

- Fig1a: Why was the line profile fitted to a Gaussian? Please additionally show the original

intensity.

- In large tissues, the PSF is typically spatially varying. How would that affect the restoration?
- The authors consistently use the term "super-resolution" for the isotropically restored deconvolution result. In the context of microscopy "super-resolution" usually refers to subdiffraction optical resolution, which is not obtained (and neither attempted) by the authors. So I think changing the term "super-resolution" to something more correct (e.g. "deconvolution") is appropriate
- How important is the cycle consistency? What would happen if only G and D^i_X are used?
- In Fig1 and Fig2, please indicate for all inset which axis is X, Y, or Z
- Why was a different blurring forward generator F used for the two CFM and lightsheet data sets? Was happens if a deep linear generator would have been used for the light sheet dataset?
- "Current LSM systems have relatively low axial resolutions mainly driven by the optic aberration that is caused by the mismatch of refractive indices between air and immersion medium [16,17]" I don't see this supported by the given references - most LSM systems use water immersion lenses that are corrected for this.

Reviewer #2 (Remarks to the Author):

This manuscript describes an axial resolution improvement approach based on the 3D cycleGAN network, for volumetric fluorescence microscopy without the need of paired 3D image stacks. The authors demonstrate the resolution enhancement capability on confocal microscopy (CFM) and open-top light-sheet microscopy (OT-LSM). The main advantage of this methods is that the network can be trained with unmatched 3D image stacks, greatly simplified training data preparation process. However, the outstanding problem is the poor accuracy of the network inferences. The artifacts / hallucinations will not provide useful information for biological studies, sometimes they are even harmful. Besides, the novelty of this methods is also limited. The resolution improvement strategy, e.g., using the lateral information to improve the axial resolution, has been previously reported by CARE (Weigert, M., Schmidt, U., Boothe, T. et al, 2018). Therefore, this manuscript is unacceptable in term of either novelty of the approach or quality of the results.

Major comments:

1. For the network structure, where the key novelty of this method is reflected, the author designed three independent discriminators both in forward and backward path to compare the XY, XZ, YZ images. Given the fact that the generator G only enhances the axial resolution for either XZ or YZ slices, the utility of the discriminators $D_{X^{(1)}}$ and $D_{X^{(2)}}$ / $D_{X^{(3)}}$, which compare the X-Y plane and YZ/XZ plane, is not clear.
2. The accuracy of the network inferences shown in Figure.1c and Supplementary Figure.3 are not convincing to me. The network outputs show noticeable artifacts, as compared to the reference images. For example, the network output in the yellow boxed regions in Figure.1c shows apparent structure loss.
3. In the resolution improvement experiments on OT-LSM (Figure. 2), the network indeed provides improved axial resolution as compared to the deconvolution results. However, quantitative measurement, such as structural similarity index (SSIM), resolution scaled error (RSE) are completely absent. However, these metrics are necessary to quantify the inference accuracy.

4. In the PSF deconvolution experiments, the network reconstruction in Figure.3a shows improved resolution but with additional noises. I doubt the similar problems will occur in CFM and OT-LSM experiments.

Minor comments

5.The computational costs on the model training and inference are not mentioned.

Reviewer #3 (Remarks to the Author):

Deep learning enables reference-free isotropic super- resolution for volumetric fluorescence microscopy

Summary: The authors present a cycleGAN-based approach for what is ultimately a super-resolution method to generate isotropic resolution output from anisotropic inputs. Since the axial (i.e. "Z") resolution of most optical systems is significantly lower than the lateral (i.e. "XY") resolution, this is a very valuable, important challenge for biological imaging, and results are delightful. The authors cleverly developed a ground-truth validation/sanity-check by rotating the sample 90 degrees to generate high xy resolution images of the sample that was previously captured in the xz plane. In spite of its imperfect nature, this effort to generate ground-truth data for validation should not be underappreciated – I cannot think of a better way to do it myself, other than imaging the sample on a more advanced microscope with isotropic resolution, which may or may not exist, and even then would still suffer from many of the same problems as their current approach (e.g. misalignment, sample degradation/bleaching, etc).

Overall, the work presents a sufficient advance for publication in Nature Communications. However, important rigorous controls and analyses are missing from the manuscript that preclude suitability for publication in its current state. Most strikingly, there is no validation of results using segmentation of actual biological structures. This omission, along with the somewhat strange fact that the only metric used is PSNR (why not SSIM?) makes me worry the results have not been sufficiently validated. Thus, performing segmentation-based analysis will be key – see for example Fang et al. 2021 Nature Methods or Weigert 2018 Nature Methods for an example of how segmentation should be used to validate model output results. In brief, segmentation of key biological structures of interest should be performed in a double-blinded fashion, then compared with bilinear interpolation, deconvolution, and ground truth data. Until biological structures of interest are validated as accurate, we cannot rest assured the output of the network is accurate enough to be biologically relevant.

NCOMMS-21-19275A

Response to the Reviewers

General Comments

We thank the reviewers for the constructive reviews. Your comments have significantly improved the revision. To address the comments carefully, we have made the following major changes:

1. We have improved our model, in terms of its performance and computational cost. We additionally considered the factor that lateral slices contain projection information from adjacent slices by the z-axis blurring. This intuition was reflected in the form of the discriminators in the forward path sampling from projection images for the generated volume in the source domain and sampling lateral slice images from the real volume in the target domain. We noticed that this change translated to meaningful improvement of the reconstruction capability of the framework. The improved performance is reported additionally in the simulation studies, the segmentation analysis via neuron tracing, and the testing on an artificially blurred volume of light-sheet microscopy (OT-LSM) images. Furthermore, in the case of an imaging where the two orthogonal axial planes (i.e. YZ and XZ) share a similar imaging condition (e.g. the confocal fluorescence imaging), one could use one discriminative network for axial sampling instead of two discriminative networks, letting one discriminative network to learn from both axial planes. This maneuver reduced the computational cost for memory by approximately 20%, further extending the ease-of-use in practice.
2. In practice, the image degradation process in the axial slices for fluorescence microscopy can be affected by a complex blend of imaging factors, many of which are nearly intractable without careful calibration and measurements. Acquiring a true ground-truth for validation was an open challenge for our task, as obtaining a truly isotropic microscopy volume in a real-world setting was nearly an impossible task for us without introducing any fundamental changes in image characteristics or resorting to super-resolution microscopy. In light of this challenge, we generated a synthetic image volume with a complex mesh network of tubular structures and simulated an image degradation process, such as a blurring in the z-axis. This allowed us to measure the effective performance of our framework in solving a physics-based inverse problem in a controlled environment. As further quantitative assessment of our framework, we also tested on an artificially blurred image from light-sheet microscopy, whose resolution was calibrated to near-isotropy. In our tests, we have learned that our model has successfully restored the degraded images much closer to the ground-truth.
3. We agree that the CARE [19] shares an important part of intuition of using the lateral slices as the target domain. However, how we embody the intuition as a deep-learning-powered solution is fundamentally different. The works in [19] are based on the supervised learning approaches, where the high-resolution lateral images are blurred using a point spread function and the neural network is trained in such pairs. Therefore, if the basic assumption of the blurring process is not correct, the performance in the real data set may be limited. Furthermore, as the supervised learning methods rely on the direct relationships between the source-target pairs, it is possible that they could be more prone to over-fitting and not suitable for generalization across a large - scale image stack. On the other hand, our method is purely an unsupervised learning approach where we do not assume any specific model for the blurring process. The difference translates to a meaningful advantage in a large-scale reconstruction, and the method can be applied in a large volume microscopy stack as shown in this paper.

4. True verification of restored details is one of most common concerns for a generative deep-learning-based method. One good news is that in contrast to the GAN (generative adversarial networks), we can prevent the model from generating artificial features in the cycleGAN based approach. This has been proven rigorously in our theoretical works [R1].

[R1]. Sim, B., Oh, G., Kim, J., Jung, C., & Ye, J. C. Optimal Transport Driven CycleGAN for Unsupervised Learning in Inverse Problems. *SIAM Journal on Imaging Sciences*, 13(4), 2281-2306, Dec., 2020

This is because cycleGAN approach can be derived as an optimal transport map between two probability distributions that can be obtained by minimizing the statistical distance between the transported probability distribution and empirical distribution in both measurement and image spaces. Specifically, as shown in the figure below, the generator G_θ transports the blur image distribution ν to synthetic high-resolution image distribution μ_θ , whereas the forward blur operator F_ϕ transports the high-resolution image distribution μ to synthetic blur image distribution ν_ϕ , and the goal of the imaging problem is to minimize the statistical distance between μ_θ and μ as well as the statistical distance between ν_ϕ and ν . If this problem is converted to a so-called Kantorovich dual formula, the cycleGAN approach can emerge. Therefore, with proper design of the network using an optimal transport framework, we can avoid generating artificial features.

Figure R1. Optimal transport geometry of CycleGAN [R1].

As illustrated in our CFM experiment and Figure 2, our method visualizes micro-structures that were not easily distinguishable from the input axial images. In our effort to validate the reconstruction capability of our framework and to further explore it, we have carried out segmentation-based analysis based on neuronal tracings of the confocal microscopy image volume. We have traced two test pyramidal neurons in the output volume using a state-of-the-art neuron tracing method, NeuroGPS-tree, followed by a human correction and verified the tracings by a slice-by-slice manner on the reference image volume, which was imaged at a 90-degree rotation. The tracing in the output volume was done blindly without any knowledge from the input image or the reference image. In our verification of the tracings on the reference image, most of the of the newly reconstructed details are shown to be real.

For details of the changes, please refer to the point-by-point responses to the reviewers in the following.

Reply to the Reviewer 1

R1C1. [Quantification] Currently, the authors quantify their methods on extended structures (brain dataset) by line plots (Fig1a,e) and the comparison against the lateral 2D slices of the 90 degree rotated reference (Fig1f). This is rather limited as it cannot describe the true 3D restoration improvement of the method. I understand that a truly isotropic 3D reference image would be impossible to obtain, but why did the authors not validate their method with a synthetically created dataset? This way, one could additionally compare against RL deconvolution in a truly quantitative way.

→ We are in total agreement that simulation will provide more rigorous examination of the framework, especially for testing a deep learning framework for solving a physics-based inverse problem. As per your recommendation, we have carried out simulation studies where we generated a synthetic volume with a randomized mesh network of tubular structures and provided evaluation of the model's performance in our latest revision. Figure R1C1 below shows a small-scale example of this. Based on this synthetic image volume as the ground-truth, we could simulate the image degradation process *in silico*: for example, axial blurring and down-sampling. Then, we trained the network of our framework on the axially blurred image volume alone. In our tests, we have learned that the network could effectively reverse the blurring process blindly, recovering the complex web-like structures in the axial plane, which were previously masked by the blurring. The improvement was consistent over different levels of blurring and down-sampling. The improvement was reported in the metrics of PSNR, SSIM and MS-SSIM. This is shown in Figure 1 of the paper.

Furthermore, in our revision, we have calibrated the OT-LSM system to provide near-isotropic resolution from the mouse brain sample. Using this as the ground-truth, we generated a *semi-synthetic* volume whose resolution is artificially degraded by the z-axis blurring. By training the network for blind deconvolution, we observed that the network provided noticeable improvement of axial image quality. The PSNR, MS-SSIM and BRISQUE metrics were reported to quantify the improvement. In our quantitative comparison with RL deconvolution based on the blurring PSF, we noticed that the RL-deconvolution has rather an adverse effect on the metrics, in comparison to the input image. For example, in case of the axial MIP images displayed in Supplementary Figure 13a, while the network output has PSNR improvement of ~3dB in comparison to the blurred input, the RL deconvolution image has a negative PSNR difference of ~1.3dB. This is likely because the RL deconvolution failed to reconstruct high frequency information that is lost from the axial blurring.

Figure R1C1 Small-scale example of the simulation process

R1C2. [Software availability] Currently the authors provide no code that would allow other researchers to use this method on their own data. Specifically, as the proposed method is unsupervised (and thus could be applied to already acquired 3D volumes) this would greatly increase the value of the paper.

→ Thank you for pointing this out. We agree that accessibility should be the key factor in developing software tools for researchers. Our framework with the main dataset is now available on Github: <https://github.com/peterpark-git/neuroclear>.

RIC3. It is worth to note, that the idea of using the lateral slices of an anisotropic microscopy volume as a target for the deconvolution of the poorer resolved axial slices is already used in the context of supervised DL deconvolution in the given reference [19]. This could be made clearer.

→ We agree that the given reference [19] shares an important part of intuition of using the lateral slices as the target domain. However, the work in [19] is based on the supervised learning approaches, where the high-resolution lateral images are blurred using a point spread function and the neural network is trained in such pairs. Therefore, if the basic assumption of the blurring process is not correct, the performance in the real data set may be limited. On the other hand, our method is purely an unsupervised learning approach where we do not assume any specific model for the blurring process. Therefore, it can be applied in a large volume microscopy stack as shown in this paper. Thanks to your suggestion, the reference to [19] is made more clearly.

RIC4. Fig. 1a: Why was the line profile fitted to a Gaussian? Please additionally show the original intensity.

→ Thank you for pointing this out. This change is reflected in our latest revision, with the original intensities in display.

RIC5. In large tissues, the PSF is typically spatially varying. How would that affect the restoration?

→ Thanks for pointing out this important question. In our work, we presented two variations of the framework to capture the image degradation process in fluorescence microscopy in the backward path of the architecture: one model with a linear generator and the other of an encoder-to-decoder structure with skip connections (U-Net). In both cases, there exist mechanisms to deal with the spatially varying PSF.

The first model addresses when the image degradation process is largely driven by the PSF convolution process, it can be approximated to a PSF convolution, which can be explicitly described and thus learned by the linear generator. In our paper, this case applies to the simulation studies, the CFM imaging, the OT-LSM imaging of fluorescence beads, and the OT-LSM imaging with artificial z-axis blurring. As the linear generator is consisted of deep convolutional layers, it is capable of encoding multiple PSF models in its latent space [R1]. As the trained network is applied in a patch-by-patch manner, the linear generator can adaptively apply the appropriate PSF convolution depending on the patch location, so it can deal with the spatially varying PSF. The explicit modeling of the blurring process using convolutional layers was previously reported in biomedical image research [R1, R2, R3] which are listed below.

[R1]. Bell-Kligler, S., Shocher, A. & Irani, M. Blind super-resolution kernel estimation using an internal-GAN. In Advances in Neural Information Processing Systems 32: Annual Conference on Neural Information Processing Systems 2019, NeurIPS 2019, December 8-14, 2019, Vancouver, BC, Canada, 284

[R2]. Sim, B., Oh, G., Kim, J., Jung, C., & Ye, J. C. Optimal Transport Driven CycleGAN for Unsupervised Learning in Inverse Problems. SIAM Journal on Imaging Sciences, 13(4), 2281-2306, Dec., 2020

[R3]. Lim, S., Park, H., Lee, S., Chang, S., Sim, B., & Ye, J. C., "CycleGAN With a Blur Kernel for Deconvolution Microscopy: Optimal Transport Geometry", IEEE Trans. on Computational Imaging, 6, 1127-1138, July, 2020.

In the second case, we used a U-Net-based architecture. With its skip connections and encoder-decoder structure, the nonlinear nature of the neural network automatically adjusts the spatially varying PSF by learning high-level representations at a semantic level. We apply this model to imaging scenarios where the image degradation process is non-linear and multi-faceted: e.g. the OT-LSM images degraded by the sample vibration from the stage drift.

RIC6. The authors consistently use the term "super-resolution" for the isotropically restored deconvolution result. In the context of microscopy "super-resolution" usually refers to subdiffraction

optical resolution, which is not obtained (and neither attempted) by the authors. So I think changing the term "super-resolution" to something more correct (e.g. "deconvolution") is appropriate.

→ Thank you for pointing this out, and we agree that it is a valid concern, as we started with a premise that we deal with microscopy images where their anisotropy is mainly attributed to the axial elongation of the blurring point-spread-function. If we could model the entire image degradation process of an imaging system to a convolution with a point spread function, its reversal process can naturally be called deconvolution. In practice, there are other factors at play that cannot be simply modeled as a convolution process with a point spread function. In our work, our additional goal was to solve other inverse problems simultaneously in addition to the blurring PSF, especially when the imaging system is not perfectly calibrated or optimized.

For example, with our confocal imaging of a mouse brain sample, we learned that the image degradation process cannot be simply be modeled as a PSF convolution. For example, we have also carried out Richard-Lucy (RL) deconvolution on the input image volume based on the PSF that is experimentally extracted. We have learned that the deconvolution does not fully improve the image quality nor reverse the entire degradation process. For example, in the figure below, the regions-of-interest (ROIs) in the bottom indicate that the network can generate intact dendritic connections where the RL-deconvolved image still visualizes the dendrites with horizontal damages. This particular artifact is likely attributed to the under-sampling in the z-axis compared to the sampling on the lateral plane.

Furthermore, our simulation studies provide another evidence. We tested the image enhancing capability of the network with a synthetic volume additionally with z-axis sub-sampling after the blurring: sub-sampling rates of 2 times, 4 times, and 6 times in our tests. The quantitative results show that the network is capable of enhancing images that were previously under-sampled, showing that our framework is capable of not only modelling a PSF response but also restoring images with high-level semantic understanding of the imaging process.

Figure R1C6 Isotropic reconstruction compared with RL-deconvolved data. a) Comparison of axial images of input, RL-deconvolution, output. b) The PSF model that is experimentally extracted.

RIC7. How important is the cycle consistency? What would happen if only G and D^i_X are used?

→ In our revision, we additionally carried out two ablation studies on the synthetic image volume where one model excludes the cycle-consistency loss and the other excludes the backward path, which is of F and D_y . For fair comparison, we used the trained network models in the same condition, such as the training iteration. The results are shown in the figure RIC7 below.

In the first case, without the cycle-consistency, we noticed that while the network performance weakened in comparison to the baseline, the image was de-blurred to some extent, shown by the improvement in the PSNR and MS-SSIM metrics. This is due to the design of the loss function still including the joint minimization of generation losses from both the blurring and deblurring. Nonetheless, as the metrics suggest, its reconstruction shown below was not as accurate as the baseline, and this confirms that the cycle-consistency contributes to improving the reconstruction accuracy. As a theoretical reference, the benefit of using cycle-consistency was previously formulated mathematically in terms of an optimal transport as an effective strategy to find the correct image transformation [R1] between two distinct domains.

[R1]. Sim, B., Oh, G., Kim, J., Jung, C., & Ye, J. C. Optimal Transport Driven CycleGAN for Unsupervised Learning in Inverse Problems. *SIAM Journal on Imaging Sciences*, 13(4), 2281-2306, Dec., 2020

In the second case, without the backward path, we noticed that the generator still managed to generate convincing visuals, whose textures were comparable to those from the comparison models. This suggests that the lateral/axial sampling of the discriminative networks in the forward path is instrumental in generating refined details. However, without the backward path, the network cannot guide its learning to revert the image degradation process and hence ends up with inaccurate representations in the latent space modeling the high-resolution manifold. This is well reflected in the inaccurate reconstructions, as suggested by the exacerbated metrics of this ablation case.

Figure RIC7 Ablation study of the framework. The networks were trained on the synthetic image volume that was blurred with a Gaussian kernel with a standard deviation of 4. PSNRs and MS-SSIMs from the test volume are reported for quantitative comparison.

RIC8. In Fig1 and Fig2, please indicate for all inset which axis is X, Y, or Z.

→ Thanks for pointing this out. The changes are reflected in our latest revision.

R1C9. Why was a different blurring forward generator F used for the two CFM and lightsheet data sets? Was happens if a deep linear generator would have been used for the light sheet dataset?

→ Thanks for pointing out this important question. In our work, we presented two variations of the framework to capture the image degradation process in fluorescence microscopy in the backward path of the architecture: one model with a linear generator and the other of an encoder-to-decoder structure with skip connections (U-Net). In both cases, there exist mechanisms to deal with the spatially varying PSF.

The first model addresses when the image degradation process is largely driven by the PSF convolution process, it can be approximated to a PSF convolution, which can be explicitly described and thus learned by the linear generator. In our paper, this case applies to the simulation studies, the CFM imaging, the OT-LSM imaging of fluorescence beads, and the OT-LSM imaging with artificial z-axis blurring. As the linear generator is consisted of deep convolutional layers, it is capable of encoding multiple PSF models in its latent space [R1]. As the trained network is applied in a patch-by-patch manner, the linear generator can adaptively apply the appropriate PSF convolution depending on the patch location, so it can deal with the spatially varying PSF. The explicit modeling of the blurring process using convolutional layers was previously reported in biomedical image research [R2, R3].

[R1]. Bell-Kligler, S., Shocher, A. & Irani, M. Blind super-resolution kernel estimation using an internal-GAN. In Advances in Neural Information Processing Systems 32: Annual Conference on Neural Information Processing Systems 2019, NeurIPS 2019, December 8-14, 2019, Vancouver, BC, Canada, 284

[R2]. Sim, B., Oh, G., Kim, J., Jung, C., & Ye, J. C. Optimal Transport Driven CycleGAN for Unsupervised Learning in Inverse Problems. SIAM Journal on Imaging Sciences, 13(4), 2281-2306, Dec., 2020

[R3]. Lim, S., Park, H., Lee, S., Chang, S., Sim, B., & Ye, J. C., "CycleGAN With a Blur Kernel for Deconvolution Microscopy: Optimal Transport Geometry", IEEE Trans. on Computational Imaging, 6, 1127-1138, July, 2020.

In the second case, we used a U-Net-based architecture. With its skip connections and encoder-decoder structure, the nonlinear nature of the neural network automatically adjusts the spatially varying PSF by learning high-level representations at a semantic level. We apply this model to imaging scenarios where the image degradation process is non-linear and multi-faceted: e.g. the OT-LSM images degraded by the sample vibration from the stage drift. In our empirical test on the OT-LSM image volume, while the restored visuals were comparable between the two variations, we noticed that the details restored by the second model were clearer. Figure R1C9 below illustrates this.

Figure R1C9 Visual comparison of network performance on OT-LSM images between the network variations. (Scalebar: 25 μm , 10 μm in ROI).

R1C10. Current LSM systems have relatively low axial resolutions mainly driven by the optic aberration that is caused by the mismatch of refractive indices between air and immersion medium [16,17]" I don't see this supported by the given references - most LSM systems use water immersion lenses that are corrected for this.

→ With our OT-LSM imaging, our main goal was to test whether our framework can restore an image where the image degradation process is governed by a mixture of multiple image degrading factors. Specifically, the OT-LSM system that we used has visual artifacts from the blurring not only by the spherical aberration caused by the aforementioned refractive index mismatch, but also other additional artifacts: for example, image-doubling artifacts from the asynchronization of the sweeping of the excitation laser and the rolling shutter of the detection sensor. Accordingly, the goal is to test whether our method can blindly learn to restore a degraded image from a *badly* calibrated system.

Additionally, in practice, water immersion lens could be an expensive option for low-budgeted microscopists. With our OT-LSM, we are testing on a scenario where the water immersion lens is missing from the system. This leads to spherical aberration from the system, causing anisotropy. The modelling of spherical aberration because of the absence of the water immersion lens is described in the figure below.

Figure R1C10 Modeling of spherical aberration in the presented OT-LSM system

Reply to the Reviewer 2

R2C1. ... the outstanding problem is the poor accuracy of the network inferences. The artifacts / hallucinations will not provide useful information for biological studies, sometimes they are even harmful.

→ As suggested, we mainly focused in our revision on measuring the accuracy of the network inferences and verifying possible artifacts or hallucinations. First, our simulations studies were designed to simulate the physical process of axial image degradation in microscopy in a biologically plausible image volume. Second, in order to verify the reconstructed biological details on a hands-on approach, we verified the neuronal tracings of pyramidal neurons with respect to the image volume that was imaged at a perpendicular angle. As changing the imaging angle affects the spatial distribution of fluorescence emission in confocal microscopy, it is still not ideal to use a rotated image volume as a reference to verify every fine detail perfectly. However, on the accuracy level of visualizing neurons that were traceable, we observed that the traced reconstructions in the network output image were

predominantly real. Third, we calibrated the OT-LSM imaging to the best of our ability to near-isotropy to provide the biological ground-truth and tested our method on the artificially blurred image.

As for the theoretical background of our method, in contrast to other GAN-based methods, our cycleGAN approach provides a more solid mathematical ground for steering away from generation of spurious features. This is mentioned in General Comment 4.

R2C2. Besides, the novelty of this methods is also limited. The resolution improvement strategy, e.g., using the lateral information to improve the axial resolution, has been previously reported by CARE (Weigert, M., Schmidt, U., Boothe, T. et al, 2018).

→ See General Comment 3 to understand the novelty of our work over CARE.

R2C3. For the network structure, where the key novelty of this method is reflected, the author designed three independent discriminators both in forward and backward path to compare the XY, XZ, YZ images. Given the fact that the generator G only enhances the axial resolution for either XZ or YZ slices, the utility of the discriminators $D_X^{(1)}$ and $D_X^{(2)}/D_X^{(3)}$, which compare the X-Y plane and YZ/XZ plane, is not clear.

→ Note that G is a 3-D generator that performs 3-dimensional deblurring of an input patch volume. As explained in the descriptions of the loss functions, the generative networks seek to maximize the loss functions and the discriminative networks seek to minimize these. The learning of G is guided in a mini-max game in this manner by the three discriminative networks, $D_X^{(1)}$, $D_X^{(2)}$, and $D_X^{(3)}$. When comparing with lateral slice images from a real image volume, $D_X^{(2)}$, and $D_X^{(3)}$ sample from the axial images of the volume generated by G , and $D_X^{(1)}$ samples from the lateral images of the same generated volume. In other words, while $D_X^{(2)}$, and $D_X^{(3)}$ seek to transform the low-resolution axial images, $D_X^{(1)}$ seeks to maintain the lateral information so that the 3D volume maintains its structural integrity without introducing any distortions in the XY plane. Therefore, we need three discriminators that compares the changes in XY, YZ, and ZX slices during the 3D deblurring/blurring steps.

R2C4. The accuracy of the network inferences shown in Figure.1c and Supplementary Figure.3 are not convincing to me. The network outputs show noticeable artifacts, as compared to the reference images. For example, the network output in the yellow boxed regions in Figure.1c shows apparent structure loss.

→ In the CFM imaging of the brain sample, we verified the network output based on the image volume that was imaged at a perpendicular angle. In practice, when an imaging angle changes as largely as 90 degrees, the excitation path and the detection path would also change accordingly, naturally also altering the spatial distribution of fluorescence emission. This means that fluorescence signals in some regions that were visualized in the input image volume could be missing in the rotated image, and vice versa. For this reason, there are no perfectly matched reference data. and hence we obtained pseudo-reference images by physically rotating the sample, and a trained network can only infer based on what is available from the input image. Therefore, the referred “artifact” in your comment is not due to the generation process, but we believe they are from the mismatch with pseudo-references.

To avoid this issue and deal with the lack of ground-truth, we performed simulation studies where the ground-truth images are known. In our simulation results, we found that our method provides accurate reconstruction without noticeable artifacts.

R2C5. In the resolution improvement experiments on OT-LSM (Figure. 2), the network indeed provides improved axial resolution as compared to the deconvolution results. However, quantitative measurement, such as structural similarity index (SSIM), resolution scaled error (RSE) are completely absent. However, these metrics are necessary to quantify the inference accuracy.

→ Thank you for pointing this out. The previous OT-LSM imaging was done in the way that the system involves many kinds of imaging artifacts and the aim of the study was to see if the trained network could alleviate those artifacts blindly. The types of the imaging artifacts are described in R1C10. As it is difficult to model the whole image degradation as a PSF convolution with a single PSF model, there is no suitable signal-processing-based method to reverse the image degradation to a level of accuracy that we can regard it as a ground-truth. For this reason, the point of comparing to the RL deconvolution results was to provide a comparison method on how to improve the image in the post-imaging stage, and computation of image quality metrics with regards to the deconvolution results would not be a good representation of the network's performance.

However, we agree that it is important to provide a quantitative analysis with the OT-LSM imaging as well. In light of this challenge, we have calibrated the LSM imaging system to the best of our capability to make it isotropic in resolution (e.g. via tight focusing of the excitation [R1]), setting it as a ground-truth. Then, in the similar manner as in the simulation studies, we artificially applied a Gaussian blurring in the Z-axis to simulate a blurring process in microscopy. Based on this ground-truth image and this artificially blurred OT-LSM image, we could carry out a quantitative analysis for blind deconvolution capability of the method in the OT-LSM system. In our revision, we reported PSNR and MS-SSIM as metrics for image quality with reference and BRISQUE as a metric for image quality without reference, which was then compared to the ground-truth. The visuals and all the metrics suggested that there is a meaningful improvement for image quality. This is shown in the Supplementary Figure S13.

[R1]. Kim, B. et al. Open-top axially swept light-sheet microscopy. *Biomedical Optics Express* (2021).

R2C6. In the PSF deconvolution experiments, the network reconstruction in Figure.3a shows improved resolution but with additional noises. I doubt the similar problems will occur in CFM and OT-LSM experiments.

→ Thank you for pointing this out. As we worked with 16-bit image formats to provide more precision to the computing process, we needed to choose the intensity window to visualize the results. When we imaged fluorescent beads and tested the imaging system to the best of its resolving capability, we needed to visualize in the intensity spectrum where the contrast is more saturated and noises are thus more visible in comparison to the fluorescent beads. In Figure 3, all the 2D images are visualized without any contrast enhancement. Thanks to your suggestion, we looked into whether the network introduces any additional noises to the image in the imaging of fluorescent beads using OT-LSM. For this analysis, we selected a small FOV containing a single fluorescent bead for comparison between input and network output. We quantified the SNR, based on the calculation method for imaging beads for PSF modeling by Wang et al. (2019) [R1]. In our analysis, the SNR metric was higher for the output image and suggest that the network did not necessarily make the image noisier. However, we did notice checker-board artifacts when the image is visualized in a higher contrast, this is likely due to the convolutional nature of the network semantically designating these regions as the background, and if necessary, there are deep-learning-based techniques [R2] to correct these artifacts in implementation.

[R1]. Wang, H. et al. Deep learning enables cross-modality super-resolution in Fluorescence microscopy. *Nature Methods* 16, 103-110 (2019)

[R2]. Odena, et al., "Deconvolution and Checkerboard Artifacts", *Distill*, 2016.
<http://doi.org/10.23915/distill.00003>

Figure R2C6 SNR calculation of a single fluorescent bead. Yellow box refers to the background ROI for SNR calculation.

SNR calculation method: Based on the method by Wang et al. (2019), the SNR was calculated as follows:

$$SNR = \frac{|s - \bar{b}|}{\sigma_b}$$

Where s is the peak value of the signal calculated from a Gaussian fit to the particle, \bar{b} is the mean value of the background ROI. σ_b is the standard deviation of the background.

R2C7. The computational costs on the model training and inference are not mentioned.

→ Thank you for pointing this out. The computational costs in the form of the number of parameters and the training time/inference are now included in the revision, at the end of *Algorithm implementation and training*.

Reply to the Reviewer 3

R3C1. Overall, the work presents a sufficient advance for publication in Nature Communications. However, important rigorous controls and analyses are missing from the manuscript that preclude suitability for publication in its current state. Most strikingly, there is no validation of results using segmentation of actual biological structures. Thus, performing segmentation-based analysis will be key – see for example Fang et al. 2021 Nature Methods or Weigert 2018 Nature Methods for an example of how segmentation should be used to validate model output results. In brief, segmentation of key biological structures of interest should be performed in a double-blinded fashion, then compared with bilinear interpolation, deconvolution, and ground truth data. Until biological structures of interest are validated as accurate, we cannot rest assured the output of the network is accurate enough to be biologically relevant.

→ Thanks for your constructive comments. We fully agree that validation of reconstructed biological reconstructions is essential to the validation of the method. As per your recommendation, we have carried out segmentation-analysis where we traced two pyramidal neurons from the confocal fluorescence microscopy image volume using the NeuroGPS-tree method and human correction and compared the results by various approaches, as shown in Figure R3C1. In visual comparison, the tracings from our network output were not limited in any direction, whereas in other methods, the NeuroGPS-tree consistently failed to trace neurites sufficiently well.

Note that by nature of this work, we purposely tested on image volumes with severe anisotropy, where the axial information is severely damaged, as shown in the confocal image of the mouse brain in Figure 2 and Figure R1C6 (shown above). While examples in Fang et al. (2021) and Weigert et al. (2018) provide a good guideline on how to evaluate segmentation results, in our test case, numeric comparison of the traced results in a brain image volume with severe anisotropy did not serve the same purpose. This is because tracing neurons in a severely anisotropic volume as our test case is not only challenging but also highly inconsistent:

the tracings between two imaging sessions on the same object differed noticeably, as illustrated in the contrast in the tracing results between the input image volume and the rotated image volume, shown in the middle column of Figure R3C1 below as well as Supplementary Figure 7. This was also the case for comparison between the input image volume and the image volume deconvolved using RL (right column of Figure R3C1). To remedy this issue to the best of our effort and resources, we measured the biological precision of the reconstruction as the metric of validating restored details by the trained network, via slice-by-slice comparison of the output tracings on the rotated image volume, thereby measuring what percentage of the traced regions actually belonged to the true positives. In our test case with the two pyramidal neurons, the output reconstructions by our method were accurate at the level of the branching of neurites and verified as biologically real on the reference image volume, with mean precision of 98.28%.

Figure R3C1 Example of tracing inconsistencies by NeuroTreeGPS with human correction, between before/after restoration by our method (left), 90 degree rotating imaging sessions (middle) and before/after post-processing using RL-deconvolution (right). RL-deconvolution was carried out using the experimentally given PSF model, with 10 iterations.

REVIEWER COMMENTS

Reviewer #1 (Remarks to the Author):

In the revised version the authors added a quantitative analysis of reconstruction quality with synthetic ground truth, clarified the exposition of the method and their relation to existing methods, and added further segmentation validation experiments and a code repository. This, together with the rebuttal, addresses the points from my review.

Small note: The demo notebook ([https://github.com/peterpark-git/neuroclear/blob/master/jupyter_notebook/Data Generator for Simulation.ipynb](https://github.com/peterpark-git/neuroclear/blob/master/jupyter_notebook/Data%20Generator%20for%20Simulation.ipynb)) still contains absolute paths specific to the authors and thus does not run. This should be changed.

Reviewer #2 (Remarks to the Author):

I appreciate the authors for their updates in improving this work, including adding more quantitative measurements about the network inference results, trying to verify the fidelity on the simulation datasets and synthetically blurred images. However, the results the authors provided are still largely insufficient to convince me with its performance advantage over alternative approaches.

1. The simulated complex mesh tubular structures used for verifying the fidelity of the network was inappropriate. The dataset looks too binarized without any noise. However, this is not the case for real-world biological imaging. Even under such an ideal condition, the network inference results still show noticeable artifacts. For example, in Figure 1.d, the network result has many artificial tubular structures, which don't exist in the ground truth. Similar artifacts can also be observed in restoration results of dendrites shown in Supplementary figure 13a. These artificial structures have raised concerns on the fidelity of the neuron tracing results shown in Figure.2c.

2. The results from real biological samples are not accurate enough as well. In Figure.3, noticeable signal loss occurs at the left corner of the 3D deconvolution results. In Figure.4c, noticeable signal loss at the middle part of the YZ plane of network outputs. Meanwhile, RL deconvolution merely shows results with quality similar to the input.

3. The authors claim many advantages over CARE, including large-scale reconstruction and no need of prior information about the degradation process. However, the authors do not provide any direct performance comparison with CARE to demonstrate their advantages regarding these points. Also, fluorescence microscopes usually have been well-calibrated and free from optical aberration. Thus, their point spread functions can be easily obtained (just like the application the author demonstrated in Figure.1-3). In this case, whether the framework proposed by the authors indeed has performance advantages over CARE and other approaches are questionable.

4. I cannot agree that the difference between the network output and reference images was from imperfect image match. The structure of network results is always different from reference images, even completely inconsistent with the input images (like the second row in Supplementary figure.8b). That's how a meaningful deep-learning image restoration model works for new, unknown structures.

5. In Figure.3a, the authors claim that there is no contrast enhancement of 2D images. Then why the 2D images in XY, XZ, YZ have different backgrounds? The author should clarify this in the manuscript and add an SNR comparison between the network output and the input images in for Figure 3.

Reviewer #3 (Remarks to the Author):

The reviewers have addressed my concerns, and the paper is now suitable for publication.

I want to add that I disagree with the majority of Reviewer 2's comments "R2C1. ... the outstanding problem is the poor accuracy of the network inferences. The artifacts / hallucinations will not provide useful information for biological studies, sometimes they are even harmful.", and "R2C2. Besides, the novelty of this methods is also limited. The resolution improvement strategy, e.g., using the lateral information to improve the axial resolution, has been previously reported by CARE (Weigert, M., Schmidt, U., Boothe, T. et al, 2018)."

These comments alone belie a misunderstanding of what was done in this paper and unwarranted hostility to the overall method.

NCOMMS-21-19275B

Response to the Reviewers

General Comments

We thank the reviewers for the constructive reviews. We appreciated not only the positive feedbacks from the reviewers but also the feedbacks on visuals of the figures, which helped us review the visualization processes for the figures and correct a minor error (Supplementary Figure 8).

To address the comments carefully, we have made the following changes in our manuscript:

- Addressed a case of possible visibility of artifacts as hallucinations when visualized as two-dimensional projection images in the discussion section of the paper.
- Addressed a case of possible deviation of local image histograms caused by the one-volume training scheme and introduced a normalization technique that addresses this issue.
- Corrected an error (two pairs of network output sub-plots and ground-truth sub-plots being switched) in the Supplementary Figure 8.

For details of the changes, please refer to the point-by-point responses to the reviewers in the following.

Reply to the Reviewer 1

RIC1. In the revised version the authors added a quantitative analysis of reconstruction quality with synthetic ground truth, clarified the exposition of the method and their relation to existing methods, and added further segmentation validation experiments and a code repository. This, together with the rebuttal, addresses the points from my review.

→ Thank for your positive feedback.

RIC2. Small note: The demo notebook (https://github.com/peterpark-git/neuroclear/blob/master/jupyter_notebook/Data_Generator_for_Simulation.ipynb) still contains absolute paths specific to the authors and thus does not run. This should be changed.

→ Thank you for your feedback. We have improved the accessibility of the software by changing the absolute path to the relative path (e.g., in the mentioned notebook, it will save a simulated image volume in the current working directory) and improved the ease of setting up the environment for running the code. So far, the software is to be tested on Linux-based systems (the original module was developed on an Ubuntu 20.04 system).

Reply to the Reviewer 2

R2C1. The simulated complex mesh tubular structures used for verifying the fidelity of the network was inappropriate. The dataset looks too binarized without any noise. However, this is not the case for real-world biological imaging. Even under such an ideal condition, the network inference results still show noticeable artifacts. For example, in Figure 1.d, the network result has many artificial tubular structures, which don't exist in the ground truth. Similar artifacts can also be observed in restoration results of dendrites shown in Supplementary figure 13a. These artificial structures have raised concerns on the fidelity of the neuron tracing results shown in Figure.2c.

→Thanks for your constructive comments. Note that the ground-truth is supposed to simulate the physical structure of a biological tissue to assess the de-convolutional and up-sampling capability of the trained network. Only after the blurring/down-sampling process, the image is considered as representing the image formulation process in the real-world biological imaging by fluorescence microscopy. Similar tubular structures have been previously used as models for simulation [R1, R2, R3]. Such complex mesh tubular structures are commonly visible as microtubules in the cytoplasm of cells or complex wirings of dendrites in the cortical region of a brain.

As per your comment on hallucinations, we investigated more deeply into possible hallucinations from a generated image volume from our simulation studies. Figure R2C1 below shows a deeper look at the mentioned Figure 1.d., for a possible case of hallucinations, which is labeled as a yellow box in the 2D (two-dimensional) MIP (maximum intensity projection) images of 20 slice depth. We assessed the validity of the zoomed-in region of 20 slice depth in both a full intensity scale and a highly saturated intensity scale (with a cut-off at 40% intensity). While a possible spurious tubular structure may be visible in the 2D MIP image of the network output, such a structure was not present nor authentic when visualized in 3D, although the region contained some scattered trace of visual artifacts only when visualized in 3D at a highly saturated intensity scale. This phenomenon is due to the nature of the network architecture: as the discriminative network learns from 2D projection images, they also may contribute to generating possible visual artifacts that are more visible when visualized as 2D projection images. This issue is successfully addressed by the generative networks with 3D deep convolutional layers, as the network also has access to the full 3D information of the given image volume and keeps false reconstructions to a negligible level in the visible intensity range. Thanks to this feedback, we added discussion on this phenomenon in our revised manuscript.

Figure R2C1 Deeper look into possible hallucinations. Figure 1d is a MIP image of 20 slice thickness that contains a possible hallucination, which is labeled as the yellow box. 3D volumes were then visualized with zoomed-in ROIs on both the full intensity scale and the saturated scale (cut-off at 40% intensity) to amplify the visibility of the possible artifact. Scale bar: 10 pixels and 2 pixels (small ROIs).

[R1]. Sage, D., Donati, L., Soulez, F., Fortun, D., Schmit, G., Seitz, A., Guet, R., Vonesch, C., & Unser, M. DeconvolutionLab2: An open-source software for deconvolution microscopy. *Methods (San Diego, Calif.)*, 115, 28–41. Feb, 2017

[R2]. Lim, S., Park, H., Lee, S., Chang, S., Sim, B., & Ye, J. C., "CycleGAN With a Blur Kernel for Deconvolution Microscopy: Optimal Transport Geometry", *IEEE Trans. on Computational Imaging*, 6, 1127-1138, July, 2020.

[R3] Qiao, C., Li, D., Guo, Y. et al. Evaluation and development of deep neural networks for image super-resolution in optical microscopy. *Nat Methods* 18, 194–202, 2021.

R2C2. The results from real biological samples are not accurate enough as well. In Figure.3, noticeable signal loss occurs at the left corner of the 3D deconvolution results. In Figure.4c, noticeable signal loss at the middle part of the YZ plane of network outputs. Meanwhile, RL deconvolution merely shows results with quality similar to the input.

Figure R2C2a. Visualization with an extended spatial range of Figure 3. The original spatial range of Figure 3. is visualized in magenta, the extended range is visualized as cyan, and the overlapping region in the border is visualized as yellow.

→ In Figure 3, the left corner of the figure does not exhibit the signal loss. As the task is to deconvolve the spherical fluorescent beads, it is reasonable that the resolved objects look as if they were shrunk into spherical objects and the signals that were present on the border region because of the PSF (point spread function) blurring would be removed after the deconvolution. To illustrate this, we re-visualized the same region of the image volume from Figure 3, this time with additional neighboring regions to show that there was no signal loss. This is shown in Figure R2C2a. above. In the border region in the left corner of the original volume, the blurred 3D objects were observed as resolved appropriately.

Figure R2C2b below shows a zoomed-in and brightened ROI of the mentioned middle part of the YZ plane of the network output. In our observation, we did not notice any signal loss: no loss of biological structural information. Figure R2C2c shows a zoomed-in ROI of the YZ plane example of RL deconvolution. We admit that the deconvolution effect by RL deconvolution was not really effective. As mentioned in the previous manuscript and the letter to the reviewers, this is because the OT-LSM system for Figure 3 was set up in the way that represents the case of a microscopy system where anisotropy is driven by additional factors other than the PSF convolution.

Figure R2C2b. Visualization with a zoomed-in ROI (yellow box) of Figure 4c. Scale: 50 μm and 10 μm (ROI). The brightness of the ROIs was increased equally both by 85% to better visualize the structure.

Figure R2C2c. Visualization with a zoomed-in ROI (yellow box) of Figure 4c. Scale: 10 μm , 2 μm (ROI)

R2C3. The authors claim many advantages over CARE, including large-scale reconstruction and no need of prior information about the degradation process. However, the authors do not provide any direct performance comparison with CARE to demonstrate their advantages regarding these points. Also, fluorescence microscopes usually have been well-calibrated and free from optical aberration. Thus, their point spread functions can be easily obtained (just like the application the author demonstrated in Figure.1-3). In this case, whether the framework proposed by the authors indeed has performance advantages over CARE and other approaches are questionable.

→ As mentioned in the manuscript, the fundamental difference of CARE from our method is that CARE requires an explicitly modeled PSF to be used for deconvolution, while our method does not require such information as priors. This difference translates to an advantage in deploying the image restoration module when we do not have access to the PSF model or are dealing with a large-scale image volume, where a single assumption of the PSF model may not be accurate. As mentioned in the original paper [R1], the isotropic restoration module of CARE relies on the assumption that the PSF is already known and constant throughout the image volume, and this assumption gets less accurate as the imaging depth of the sample tissue increases. Thus, less accurate assumption of priors for CARE may negatively affect its performance.

As per your suggestion, we have applied CARE to the simulated image volume that was processed with a Gaussian blurring with a standard deviation of 4, which is the baseline condition for our simulation experiment. Simulation is ideal for application of CARE as the PSF model was already precisely modeled. The application of CARE was carried out in the docker environment and using the Jupyter notebook, both of which were provided by the original authors, so that the exact pipeline is used to process the input volume. As the blurring PSF is already known, the same PSF was used for CARE.

Figure R2C3 below shows the comparison of our method with CARE. In our test, CARE did not accurately reconstruct the complex mesh structures; there was a significant loss of structural information in the axial images, especially along the X-axis and Y-axis. While the axial output images by CARE became less blurry in the visuals in comparison to the input image, its metric improvement was also lower than that of our proposed method. The metrics were calculated on the same test volume of 700x700x700 pixels. The CARE output showed 17.73 dB of PSNR and 0.74 of MS-SSIM, versus 19.03 dB and 0.86 from the output of the proposed method.

In our analysis, the CARE module may have two issues with its design. First, the data-pairing approach for supervised learning by CARE does not properly convey the domain difference between low-resolution axial images and high-resolution lateral images. CARE trains the network using matched pairs of 2D snapshots where the target images are from high-resolution lateral images and the source images are from simulated low-resolution axial images. Here, the simulated axial images, as the source images, are acquired by applying the anisotropic transformation (PSF convolution and sub-sampling) to the lateral images. The issue here is that the lateral images, of an already blurred image volume, contain projections from adjacent slices by the z-depth blurring. Applying the anisotropic transformation to these slices does not accurately simulate the axial blurring. This design would make sense if we apply the anisotropic transformation to lateral slices of

the ground-truth volume: axial blurring to an isotropic image volume. Therefore, this pairing does not accurately describe the domain difference between axial images and lateral images and may not accurately guide the network to learn the transformation between the domains. Second, as trained with 2D matched pairs of 2D snapshots, the CARE module has access to only 2D information for the learning and the inference of the model. This lack of access to the complete 3D information prevents the model from learning the coherence of the 3D structures.

Figure R2C3. Comparison of the proposed method with CARE. (a) MIP images with 20 slice depth. (b) 3D visualizations with PSNR and MS-SSIM as metrics for the entire test volume 700x700x700 pixels. Scale: (a) 20 pixel, 5 pixel (ROI), and (b) 10 pixel

[R1]. Weigert, M., Schmidt, U., Boothe, T. et al. Content-aware image restoration: pushing the limits of fluorescence microscopy. Nat Methods 15, 1090–1097, 2018.

R2C4. I cannot agree that the difference between the network output and reference images was from imperfect image match. The structure of network results is always different from reference images, even completely inconsistent with the input images (like the second row in Supplementary figure.8b). That's how a meaningful deep-learning image restoration model works for new, unknown structures.

→ As mentioned in the first revision, the reference images were acquired by imaging the brain sample by physically rotating at 90 degrees and registering the image spaces between the two image sessions to a cellular level (by applying affine transformation with registered anatomical landmarks). As the excitation path of the confocal microscopy will run through the sample at a perpendicular angle, this configuration means that the spatial distribution of fluorescence emission, especially in a large-scale volume, across the image space will be different to a noticeable degree. This leads to some parts of fluorescent objects get excited differently and thus may not exhibit the pixel-perfect one-to-one matching in their visuals between the imaging sessions.

However, your comment on the Supplementary Figure 8b encouraged us to review the visualization of the figure, and we noticed that we made an error of swapping the output and the reference image pair in two cases (shown in the Figure R2C4 below). Thanks to your comment, we have corrected this error in our latest revision.

Figure R2C4 the corrected version of Supplementary Figure 8 with mark-ups on swapped pairs.

R2C5. In Figure.3a, the authors claim that there is no contrast enhancement of 2D images. Then why the 2D images in XY, XZ, YZ have different backgrounds? The author should clarify this in the manuscript and add an SNR comparison between the network output and the input images in for Figure 3.

→ Thank you for pointing this out. We reviewed the visualization process for Figure 3. The signal range of the overall input volume, which spans $360 \times 360 \times 160 \mu\text{m}$ with a voxel size of $0.5 \mu\text{m}$ and measured in 16-bit, was initially scaled with respect to the global minimum and maximum. This scaling made the PSF bead example in Figure 3 relatively dimly lit, in comparison to the global maximum (the fluorophore that emits the maximum amount of fluorescence in the volume). For this reason, we select a linear visualization window based on the minimum and maximum of an input image of the selected region to visualize all other relevant images. The output slice images in the previous revision were visualized in each orthogonal plane with respect to the signal ranges of the corresponding input slice images. Additionally, because of the design of the OT-LSM [R1], which has uneven imaging formation between imaging axes, it is possible that the system could introduce slight differences to the intensity profiles between XY, XZ, and YZ, but the visible differences should be at a negligible level, as shown by the 3D visualizations from Figure 3; otherwise, the orthogonal planes will be distinguishable.

With this premise, no contrast enhancement was made, in the sense that the output images were not enhanced nor altered by any means in comparison to the input images. Thanks to your feedback, we visualized all slice images according to a single visualization scale, based on the maximum and minimum value of the input image. This is illustrated in the figure below, Figure R2C5. More importantly, this was never reflected in our metric calculations: we used *raw intensity values* in our calculations of this metric. In the PSF deconvolution experiment, the assessment metric primarily measures the model's capability to deconvolve and resolve the fluorescence beads and thus is described by how closely axial FWHM values of the generated image were estimated to the lateral FWHM values.

We also investigated how SNR (Signal-to-noise ratio) changes before and after the image restoration. However, defining SNR here is not simple, as we do not have access to the matching noise-free images for our data. We instead used the definition by Wang et al. (2019) [R2], as follows:

$$SNR = \frac{|s - \bar{b}|}{\sigma_b}$$

where s is the peak value of the signal calculated from a Gaussian fit to the particle, \bar{b} is the mean value of the background ROI. σ_b is the standard deviation of the background.

This definition means that SNR values would correlate with additive noise in images but also will be affected by how images are normalized. In our test, the normalization of the output images was important for a fair comparison of SNR. For the SNR comparison, we calculated for more than 300 bead objects. The raw network output images, without any post-processing, had lower SNR values than the input images. This difference was mainly due to peak values being noticeably lower for the raw network output images. After being normalized to the histograms of the input images, the network output images had consistently higher SNR values. As per your comment, we added this into our discussion to address the importance of proper post-processing of network output data, along with Supplementary Figure 17.

Figure R2C5 Visualization of the raw 16-bit data for the PSF bead example in Figure 3. (a) the raw 16-bit images in Figure 3 were visualized in an equal manner for all slice images according to a single visualization scale. (b) the visualization range of the images was selected based on the minimum and maximum of the input image of the selected region-of-interest for Figure 3.

[R1]. Kim, B. et al. Open-top axially swept light-sheet microscopy. *Biomedical Optics Express* (2021).

[R2]. Wang, H. et al. Deep learning enables cross-modality super-resolution in Fluorescence microscopy. *Nature Methods* 16, 103-110 (2019)

Reply to the Reviewer 3

R3C1. The reviewers have addressed my concerns, and the paper is now suitable for publication. I want to add that I disagree with the majority of Reviewer 2's comments

"R2C1. ... the outstanding problem is the poor accuracy of the network inferences. The artifacts/hallucinations will not provide useful information for biological studies, sometimes they are even harmful.", and "R2C2. Besides, the novelty of this methods is also limited. The resolution improvement strategy, e.g., using the lateral information to improve the axial resolution, has been previously reported by CARE (Weigert, M., Schmidt, U., Boothe, T. et al, 2018)."

These comments alone belie a misunderstanding of what was done in this paper and unwarranted hostility to the overall method.

→ We sincerely appreciate your positive feedback.

REVIEWERS' COMMENTS

Reviewer #2 (Remarks to the Author):

I appreciate the authors for their updates in improving this work, including the explanation about the artifacts, comparing with CARE, and exploring more applications. These additions partially address my previous concerns. However the CARE results shown in the comparison don't make sense. They look too poor and obviously are not consistent with the results shown in the original CARE paper (Fig.3). Overall, while I agree that this manuscript might show a few improvements as compared to the previous approaches, it does not show convincing conceptual advances and performance advantages.

I noticed that other reviewers did not raise similar concerns, so whether the significance and impact of this work qualified for Nature Communications is left to the discretion of the editorial team of Nature Communications.

NCOMMS-21-19275C

Response to the Reviewers

General Comments

We thank the reviewer #2 for the constructive feedback.

Reply to the Reviewer 2

R2C1. I appreciate the authors for their updates in improving this work, including the explanation about the artifacts, comparing with CARE, and exploring more applications. These additions partially address my previous concerns. However, the CARE results shown in the comparison don't make sense. They look too poor and obviously are not consistent with the results shown in the original CARE paper (Fig.3). Overall, while I agree that this manuscript might show a few improvements as compared to the previous approaches, it does not show convincing conceptual advances and performance advantages. I noticed that other reviewers did not raise similar concerns, so whether the significance and impact of this work qualified for Nature Communications is left to the discretion of the editorial team of Nature Communications.

→ We appreciate your constructive feedback. With regard to the exacerbated performance of CARE [R1] for our simulation dataset, the difference in performance can be attributed to fundamental differences in structural characteristics between our simulation dataset and the CARE dataset and the sampling strategies between our method and CARE. Our simulation dataset contains very dense webs of three-dimensional (3D) tubular structures, which were randomized such that they do not exhibit any discernible global patterns in slice views. Figure R1C1 below illustrates this feature. This implies that their 3D tubular geometries are not well described by slice views. As CARE samples in a slice-by-slice manner and employs two-dimensional (2D) generative networks, this sampling strategy could negatively affect the performance for our dataset. Our method addresses this issue by sampling for the discriminative networks by taking projection images, which better describe the 3D geometrical structures, and employing 3D generative networks.

Figure R1C1: Slice view (left) and projection view with 30 slice-depth (right) of the ground-truth image for the simulation

On the other hand, the CARE dataset (Fig. 3 in [R1]) contains biological structures that are discernible as global patterns sufficiently in slice views: e.g., ellipse-shaped nuclei from a developing zebrafish (*Danio rerio*) eye. Therefore, the 2D learning scheme of CARE could be very well capable of learning global patterns for images where image slices alone contain most of salient structural information to reconstruct from the degraded images. However, this difference in performance does not necessarily indicate that our method will be superior to CARE in every practical scenario. It is yet unclear how exactly the geometrical property of the imaged sample is relevant to the sampling strategy for the neural network training, and this could be an interesting topic for future research.

[R1]. Weigert, M., Schmidt, U., Boothe, T. et al. Content-aware image restoration: pushing the limits of fluorescence microscopy. *Nat Methods* 15, 1090–1097, 2018.